# Oncogenic mutant RAS signaling activity is rescaled by the ERK/MAPK pathway

Taryn E Gillies[1,†], Michael Pargett[1,†], Jillian M Silva[2], Carolyn K Teragawa[1], Frank McCormick[2,3] & John G Albeck[1,*]

## Abstract

**Activating mutations in RAS are present in ∼ 30% of human tumors, and the resulting aberrations in ERK/MAPK signaling play a central role in oncogenesis. However, the form of these signaling changes is uncertain, with activating RAS mutants linked to both increased and decreased ERK activation *in vivo*. Rationally targeting the kinase activity of this pathway requires clarification of the quantitative effects of RAS mutations. Here, we use live-cell imaging in cells expressing only one RAS isoform to quantify ERK activity with a new level of accuracy. We find that despite large differences in their biochemical activity, mutant KRAS isoforms within cells have similar ranges of ERK output. We identify roles for pathway-level effects, including variation in feedback strength and feedforward modulation of phosphatase activity, that act to rescale pathway sensitivity, ultimately resisting changes in the dynamic range of ERK activity while preserving responsiveness to growth factor stimuli. Our results reconcile seemingly inconsistent reports within the literature and imply that the signaling changes induced by RAS mutations early in oncogenesis are subtle.**

**Keywords** computational modeling; epidermal growth factor; FRET biosensor; RAS disease; single-cell kinetics

**Subject Categories** Cancer; Signal Transduction

**Mol Syst Biol. (2020) 16: e9518**

## Introduction

The RAS GTPases act as molecular switches, alternating between an inactive GDP-bound state and an active GTP-bound state. In the active state, RAS proteins have a greatly increased binding affinity for their effectors (Gremer *et al*, 2011), which in mammalian cells drive multiple cell growth signaling pathways. The net signaling activity of RAS in the cell represents a balance between two classes of proteins: GTPase-activating proteins (GAPs), which inactivate RAS by increasing its GTPase activity, and guanine nucleotide exchange factors (GEFs), which catalyze the dissociation of GDP and return RAS to the active GTP-bound state. Though RAS proteins are considered binary switches on the molecular level, the collective behavior of the thousands of RAS proteins present inside each cell is analog in nature. The relative activity of GAPs vs. GEFs in the cell determines the fraction of RAS molecules in the active state, which in turn regulates the activity of downstream processes.

RAS mutations occur frequently in cancers, especially those of the pancreas, lung, or colon (Fernandez-Medarde & Santos, 2011), and typically have the effect of increasing the signaling output of one of the RAS isoforms. Most oncogenic RAS mutations (85%) occur in the KRAS isoform, with 11% in NRAS and 4% in HRAS (An & Harper, 2018). Across all isoforms, 98% of oncogenic mutations are located at G12, G13, and Q61 (Prior *et al*, 2012) and render the RAS proteins GAP-insensitive to varying degrees. The net effect of these mutations is to increase the fraction of RAS proteins in the GTP-bound active state, which enhances their binding affinity to effectors, including the RAF kinases (Gremer *et al*, 2011; Smith *et al*, 2013; Hunter *et al*, 2015). RAF initiates a kinase cascade involving MEK and ERK (Fig 1A), which plays a primary role in tumor development and is a pharmacological target for cancer therapy. ERK phosphorylates hundreds of downstream targets (Yoon & Seger, 2006), many of which are transcription factors controlling cell cycle progression and cell migration.

While RAS mutations are widely thought to initiate tumors by enhancing the activity of RAF/MEK/ERK signaling to drive tumorigenic cellular behaviors, this model is not consistent with all of the data available. A number of observations deviate from this simple linear view of RAS signaling. First, the observed frequency of RAS mutations in cancer does not correlate with the strength of their effect on RAS GTPase activity. Mutations of intermediate strength are most prevalent, and the strongest mutations are found infrequently (Li *et al*, 2018). Second, the mutational status of RAS is poorly correlated with average levels of active dually-phosphorylated ERK (ppERK) both in tumor cell lines (Omerovic *et al*, 2008; Yeh *et al*, 2009) and in genetically engineered mouse models. In fact, converting a wild-type *Kras* gene to an activating mutant can actually *reduce* average ppERK levels, despite inducing tumor formation (Tuveson *et al*, 2004). These data contrast with

---

1 Department of Molecular and Cellular Biology, University of California, Davis, CA, USA
2 UCSF Helen Diller Family Comprehensive Cancer Center, San Francisco, CA, USA
3 Frederick National Laboratory for Cancer Research, Frederick, MD, USA
*Corresponding author. E-mail: jgalbeck@ucdavis.edu
†These authors are contributed equally to this work.

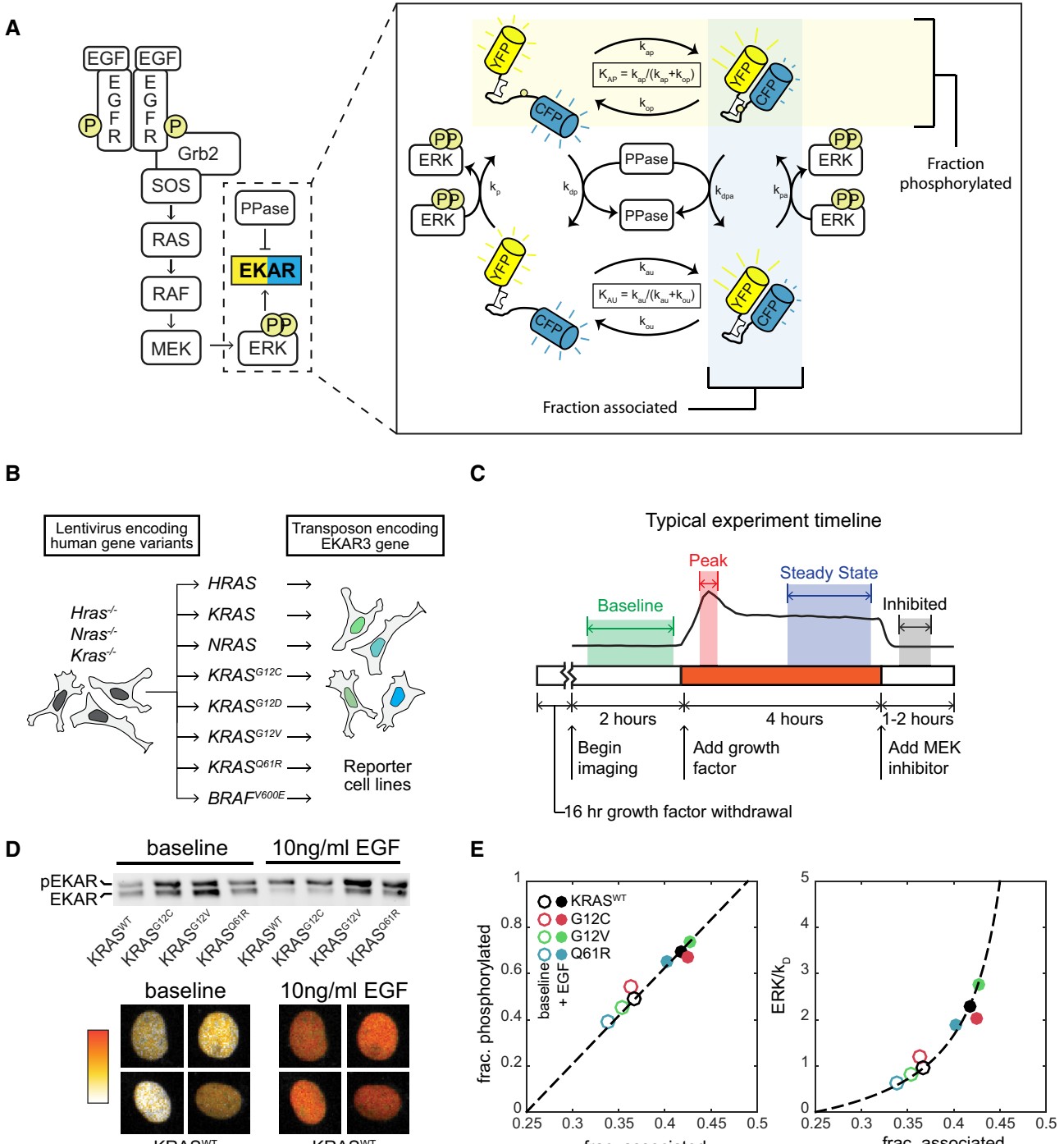

**Figure 1. Platform for ERK activity measurement in MEF cell lines expressing a single RAS isoform.**

A   Schematic of EGF signaling through RAS to ERK, including the EKAR3 sensor. The detail image at right depicts the cycle of EKAR3 phosphorylation by ERK, binding and unbinding of the internal WW domain to the phosphorylated threonine residue, and removal of the phosphate by phosphatases. Spontaneous association of the fluorophores in the absence of phosphorylation contributes to background signal and is included in activity calculations.

B   Construction scheme for cell lines bearing a single RAS isoform, using H/K/N-RAS knockouts.

C   Diagram of the typical experiment timeline. Shaded regions indicate time windows that are averaged for each measurement.

D   Sample calibration data for the EKAR3 reporter, consisting of Phos-Tag immunoblot for phospho-EKAR (upper) and live-cell imaging of reporter FRET activity (lower) under matched conditions for 4 cell lines that span the full range of ERK activity levels. Ratiometric images of four individual nuclei from the KRAS^WT line, which show the largest change from baseline to peak, are shown before and after stimulus as a representative example of the image data.

E   Calibration curves for ERK activity. Fraction of EKAR3 phosphorylated is shown vs. the fraction in the associated conformation by FRET (left). The ERK to phosphatase activity ratio (right) is derived from a model of EKAR3 (see Appendix Supplementary Methods). Each marker represents the mean value from one cell line with (filled circle) or without (open circle) EGF treatment, from 3 replicate live-cell samples and 4 replicate immuno blot samples.

observations that ectopic expression of RAS mutants does drive strong over-activation of ERK, as would be expected from simple amplification by RAF/MEK/ERK (Park *et al*, 2006; Konishi *et al*, 2007).

These contradictory observations could arise for various reasons, including feedback in the RAS/ERK pathway (Courtois-Cox *et al*, 2006), oncogene-induced senescence (Sarkisian *et al*, 2007), additional mutations, or tissue- or cell type-specific effects (Brandt *et al*, 2019). However, these possibilities cannot be disentangled without first addressing a major technical limitation inherent in the immunoblots and kinase assays that have been used almost exclusively to date. Because immunoblot measurements are relative, not absolute, it is not typically possible to compare the magnitude of ERK activation across datasets or studies. Furthermore, ERK activation has been shown to be pulsatile and heterogeneous (Albeck *et al*, 2013; Regot *et al*, 2014), so the relative differences observed when blotting for active ERK could have multiple biochemical interpretations: More ERK proteins may be active per cell, or cells with active ERK may be more frequent in the population (Birtwistle *et al*, 2012; Purvis & Lahav, 2013). Increases in the magnitude, frequency, or duration of ERK activation pulses would all yield the same result via immunoblot, though each of these signaling changes would imply different effects on gene expression and warrant different approaches for pathway directed therapy.

To clearly distinguish the forms of ERK activity that result from RAS mutations, we combined live-cell and immunoblot techniques to study a panel of cell lines each expressing only one wild type or mutant isoform of human RAS in an isogenic background. To unequivocally measure ERK activity, we employed a genetically encoded Förster resonance energy transfer (FRET)-based sensor (EKAR3) and calibrated it to deliver a quantitative linear readout of ERK substrate phosphorylation. The live-cell sensor allows measurement of cell-to-cell heterogeneity and signaling dynamics for a more detailed view of ERK activity at the cellular level. Complementing live-cell data with immunoblot measurements of RAS/ERK pathway components and computational modeling, we found that ERK activity is strikingly constrained in cells expressing mutant KRAS. When unstimulated, KRAS mutant cells exhibit only moderately elevated ERK activity compared to the wild type, and when stimulated reach peak activity no greater than the wild type. These findings outline a new unified model for how elevated RAS activity is modulated by downstream effectors and for which signaling characteristics may be relevant in cancer.

## Results

### A platform to quantify ERK activity downstream of individual RAS isoforms

To evaluate the cellular signaling capacity of each RAS isoform individually, we utilized a panel of genetically engineered mouse embryonic fibroblasts (MEFs) in which the genes for the three major RAS isoforms (*Hras, Kras,* and *Nras*) have been functionally deleted and complemented with a single constitutively expressed human cDNA (Drosten *et al*, 2010; Fig 1B). Human proteins expressed are: HRAS, KRAS, NRAS, KRAS[G12C], KRAS[G12D], KRAS[G12V], KRAS[Q61R], or the oncogenic RAF gene BRAF[V600E], in which case no RAS isoform

is expressed. In these cell lines, the signaling behavior of each RAS protein isoform can be characterized in isolation both from other isoforms and from locus-specific variations in transcriptional regulation. To track the resulting signaling activity with high temporal resolution, we transfected each MEF cell line with EKAR3, a live-cell FRET-based ERK activity reporter (Harvey *et al*, 2008; Sparta *et al*, 2015). EKAR3 is directly phosphorylated by ERK, acting as a synthetic substrate. Intramolecular binding of the reporter's WW domain to the phosphorylated residue in the substrate domain induces a FRET interaction that can be visualized by observing changes in the CFP/YFP ratio using time-lapse fluorescence microscopy. This interaction is reversible by phosphatases, allowing the reporter to indicate transient changes in the ERK:phosphatase activity ratio (Fig 1A). The resulting imaging data were analyzed with a custom image analysis pipeline (see Materials and Methods), typically yielding 100–300 single-cell time series measurements of ERK activity from each replicate of an experimental condition.

To enable accurate comparisons between the single RAS cell lines, we developed a workflow to make quantitative live-cell measurements of ERK activity (Fig 1C). Signaling activity in the absence of external stimulation, a condition we term "baseline", was quantified in cells cultured with neither serum nor growth factors for at least 16 h prior to imaging. Responses to receptor stimulation were quantified by introducing growth factor after several hours of baseline imaging. As a negative control for ERK reporter measurements, we treated cells with the highly specific MEK inhibitor PD0325901 (MEKi), which rapidly inhibited the EKAR3 signal in all cell lines. In all live-cell experiments, a 100 nM MEKi treatment was applied just prior to ending the experiment; this measures the cell-specific residual EKAR3 signal, accounting for non-specific fluorescence. The signal from the EKAR3 reporter was derived from the intensity ratio of the cyan and yellow fluorescent channels (CFP/YFP) and corrected for background as well as excitation and filter spectra. The corrected EKAR3 signal linearly reflects the fraction of reporter molecules in a FRET conformation, which is in turn linearly related to the fraction of molecules phosphorylated by ERK (Birtwistle *et al*, 2011). To calibrate, we used Phos-Tag immunoblotting to quantify the fraction of the EKAR3 reporter that is phosphorylated in various samples and conditions (Fig 1D) and fit these values against the average corrected EKAR3 signal for the same cell lines and conditions. For this calibration experiment, we selected conditions which capture a wide range of ERK activity, including the highest observed ERK activity measurement in KRAS[WT] stimulated with 10 ng/ml EGF. In concert with a mass action model of substrate phosphorylation, this calibration yields a linear measure of the ERK:phosphatase ratio (Fig 1E), i.e., the concentration of active ERK divided by the concentration of any active phosphatases that dephosphorylate the reporter (see Appendix Supplementary Methods for details). As these competing phosphatases also presumably act on endogenous ERK targets, the ERK activity measurement reflects not just the levels of active ERK, but the *net* effect of ERK on its substrates. These data confirm that our ERK activity measurement remains linear across the ERK activity ranges investigated in this study.

To demonstrate the utility of this platform for assessing inhibitor activity, we treated the panel of reporter cells with ARS-853, an inhibitor specific to KRAS[G12C]. Following treatment with ARS-853, ERK activity decreased over the course of 60 min in KRAS[G12C] MEFs, but

not in any of the other KRAS cell lines (Fig 2A). Thus, allele-specific drug responses can be identified and quantified using the reporter cell panel. Furthermore, because ARS-853 inhibits the only KRAS isoform present in KRAS[G12C] cells, we used this condition to estimate the RAS-independent background level of ERK activity. Following ARS-853 treatment, EKAR3 signal decreased to a level approximately equivalent to that of untreated KRAS[WT], followed by a small rebound. This similarity suggests that the ERK activity contributed by RAS-independent sources is near the minimal baseline value.

We next used our platform to perform a comprehensive survey of the effects of various growth factors on each cell line. We stimulated the MEF cell line panel with six growth factors known to activate RAS/ERK signaling: EGF, IGF, FGF, HGF, PDGF, and amphiregulin. Two to three concentrations of each growth factor were tested across three biological replicates, yielding activity "traces" from approximately 400 cells per condition (Fig 2B). ERK kinetics differed depending on the growth factor. For example, FGF induced sustained ERK activity without pulsatile behavior, while

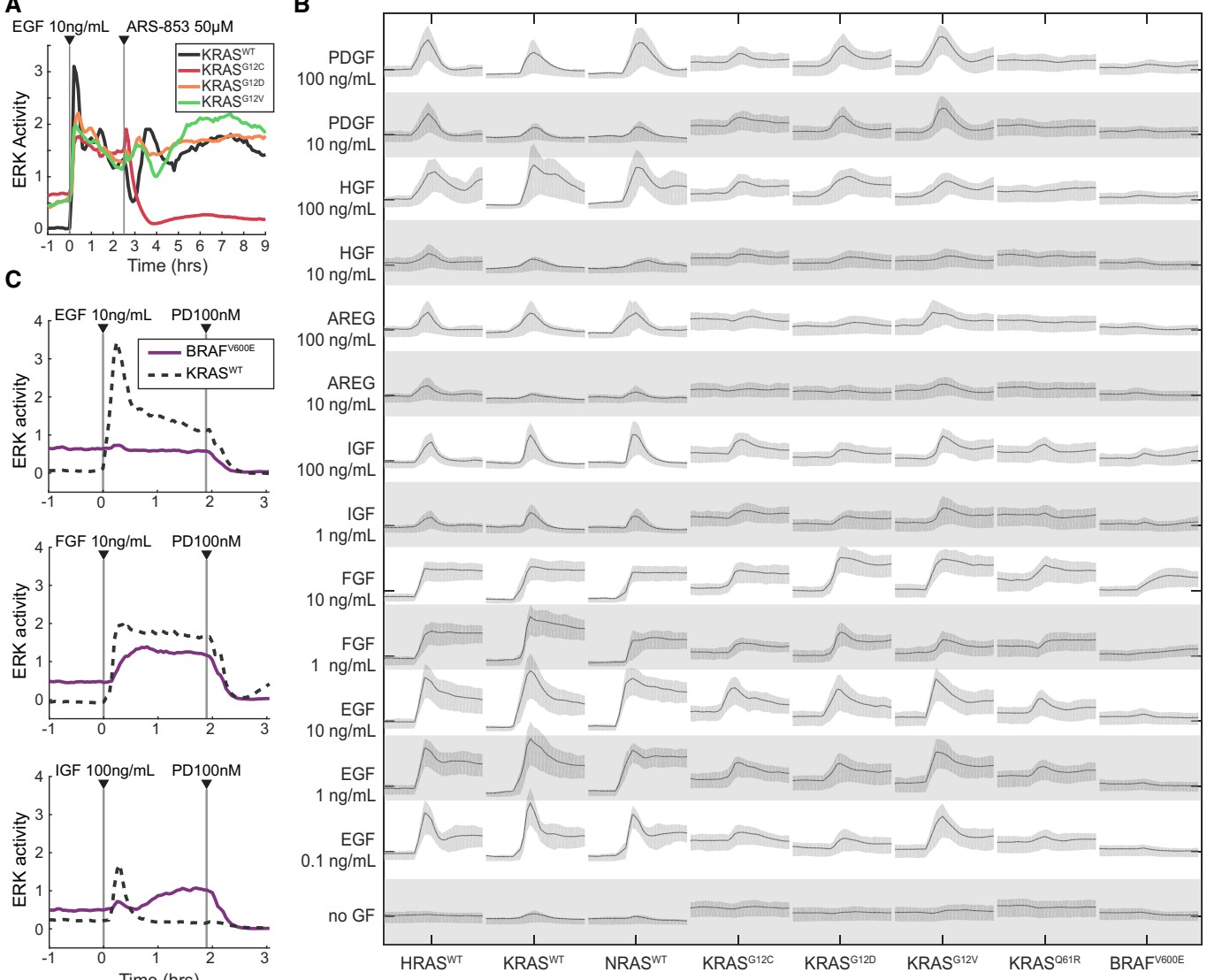

**Figure 2. Activity profiles of MEF cell lines expressing a single RAS isoform.**

A  Demonstration of the system measuring a cell line-specific response via ARS-853, a RAS activity inhibitor specific to the KRAS[G12C] mutant. Traces are median values from a representative experiment. Experiment was replicated 3 times.

B  Graphical summary of single RAS isoform cell lines (labeled along bottom) stimulated by a panel of growth factors (labeled along left). Each panel of the matrix shows the time series of ERK activity with the indicated growth factor spiked in after beginning imaging. All scales are equal; *x*-axis: time; *y*-axis: ERK activity. Lines indicate median of single-cell measurements over time, and shaded regions denote the 25th–75th percentile region, across 3 replicate cultures (6 for no GF).

C  Demonstration of RAS-independent activity from ligands other than EGF, evidenced by response in the BRAF[V600E] cell line lacking H/K/N-RAS. Traces are median values from a representative experiment. Experiment was replicated 5 times.

Source data are available online for this figure.

IGF induced a single ERK activity pulse, approximately 30–40 min in duration, immediately following stimulation. The BRAF[V600E] cell line showed moderate baseline ERK activity, consistent with previous studies in which this allele was expressed in MEFs at endogenous levels (Mercer *et al*, 2005), but it was not expected to respond to any growth factor stimulus because it lacked all RAS genes. However, both FGF and IGF induced elevated ERK activity in BRAF[V600E] cells (Fig 2C), indicating an ERK response that is not mediated via HRAS, KRAS, or NRAS, which could occur through other GTPases with the potential to activate RAF, such as RRAS or RAP1. By contrast, EGF induced high amplitude ERK activity in both mutant and wild-type RAS cells, without evidence of H/K/NRAS-independent activity in the BRAF[V600E] cell line. The remaining growth factors, PDGF, HGF, and amphiregulin, did not induce activity in BRAF[V600E] cells, but induced weaker or more transient ERK responses than did EGF in RAS-expressing cells. We therefore focus on EGF for the bulk of our subsequent analysis, because it induces a strong ERK response without H/K/NRAS-independent effects.

### RAS mutants only moderately elevate ERK activity, and only without stimulation

With quantitative single-cell resolution available, we addressed the question of how the RAS protein isoforms differ in their ERK activity patterns. After growth factor withdrawal, cells were stimulated with either media alone, or media with EGF to a final concentration of 10 ng/ml (Fig 3A–C). Across all cell lines, EGF stimulation initiated a rapid ERK activity peak ~ 15 min after stimulation, followed by attenuation over 1.5–2 h to reach a steady-state level (Fig 3B and C), with HRAS[WT] and NRAS[WT] cells exhibiting slower attenuation than any of the KRAS isoforms. Responses in single cells were qualitatively similar to the average, though each cell showed variation over time (Fig 3C). To statistically compare responses across single cells, we decomposed each single-cell ERK trace into parameters: average baseline activity, peak stimulated activity, stimulated amplitude, and average steady-state activity 2 h after stimulation. To compare the tendency for sporadic and time-varying activity, we sought a metric similar to the coefficient of variation (CV). However, when used on time series data, the CV neglects time and only reflects how far samples deviate from the mean regardless of when they occurred. We instead compute a metric we term "volatility". This is calculated by first differentiating the ERK activity per cell, then taking the absolute value and averaging over the time series. As with the CV, we scale volatility by the mean value for that cell. This metric is the time-dependent equivalent of the CV in that it is the mean-scaled average of deviations from the past time point, where the CV is the mean-scaled average of deviations from the mean. Low volatility indicates flatter more consistent activity, while higher volatility indicates more pulses, or changes in activity level over time.

Baseline ERK activity was detectable in all cells but varied in magnitude among the RAS isoforms (Fig 3D). All KRAS mutant cell lines, as well as HRAS[WT], exhibited significantly elevated baseline ERK activity compared to KRAS[WT], with the highest levels observed in KRAS[Q61R] and KRAS[G12C]. Mutant KRAS and BRAF[V600E] cells were less volatile over time than KRAS[WT] cells under baseline conditions (Fig 3E). Minimal differences in volatility were detected between HRAS[WT], KRAS[WT], and NRAS[WT] cells (Fig 3E). The higher baseline

activity in mutant KRAS isoforms compared to wild type is qualitatively consistent with constitutively higher GTP loading of these GTPase-deficient RAS proteins, which would also reduce variability in RAS activation by obscuring minor spontaneous activation events, such as autocrine signals.

The quantitative parameters of the ERK response to growth factor stimulation, including the amplitude and duration of activity, play an important part in shaping downstream cellular responses (Ebisuya *et al*, 2005; Nakakuki *et al*, 2010). We therefore explored the differences between these parameters in mutant and wild-type KRAS cells. While the rise and fall of ERK activity occurred with similar kinetics across all KRAS variants, the average peak ERK activity in the KRAS mutant cell lines was unexpectedly equal to or lower than KRAS[WT] (Fig 3B). However, differences in the average ERK activity could result from heterogeneity between cells, and upon examination, we found that the percentage of cells with a detectable ERK response to EGF was significantly reduced in KRAS mutant lines (Fig 3F and G). KRAS[Q61R] cells in particular exhibited drastically reduced response rates. This reduced response could arise from a functionally resistant subpopulation of cells, but could also result from the difficulty of detecting smaller amplitude responses in cells with elevated baseline activity. Therefore, to validate the response measurement, we examined the correlation of response frequency with baseline activity. While the response rate does vary with average baseline activity for most mutants (Fig 3H), correlation at the single-cell level is quite poor for all cell lines (Fig 3I and J); many high baseline cells clearly respond and many low baseline cells do not. Thus, the population-averaged peak ERK activity is genuinely reduced in KRAS mutant cells by a lower probability of response for each cell.

To remove the bias introduced by non-responding cells and more accurately compare average ERK responses, we filtered the ERK activity dataset to include only cells with a distinguishable response (Fig 3K–M). In this filtered dataset, the peak ERK responses in KRAS mutant cells were still equivalent to or less than those of the KRAS[WT] cells (Fig 3K). Kinetics of the growth factor response also remained similar after correction for non-responding cells (Fig 3L and M). The only distinction observed was that steady-state ERK activity 2 h after stimulation was higher in NRAS[WT] and KRAS[G12V] cells, compared with KRAS[WT], implying a slower attenuation (Fig 3L). Thus, even accounting for a reduced frequency of response, KRAS mutant cells exhibit peak ERK activity no higher than wild-type lines and show no other notable differences. The similarity of peak ERK responses across mutants is unexpected given the range of RAS GTPase activities represented in this panel of cell lines and implies that the upper limit of ERK activity is subject to tight regulation. Altogether, when individual cell variability is accounted for, the only broadly consistent distinction between ERK activity in KRAS[WT] and mutant isoforms in this system is moderate elevation of unstimulated activity.

### Immediate feedback from ERK is distributed and relatively weak in RAS mutants

The RAS-ERK pathway is subject to multiple feedback effects triggered by ERK activity that could account for the strict moderation of mutant RAS signaling. We therefore assessed the involvement of these feedback loops by comparing activity at several points in

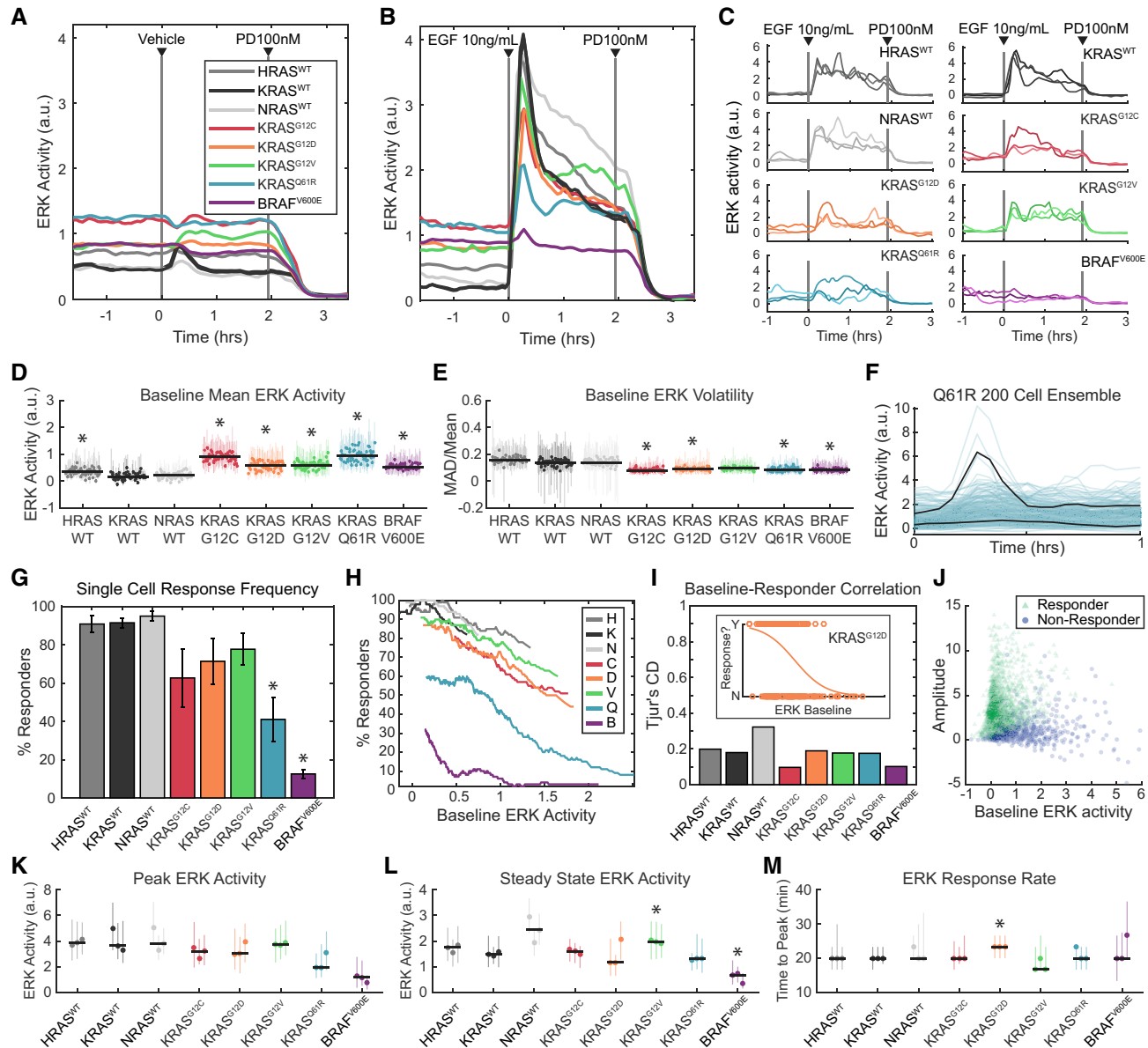

**Figure 3. Kinetic analysis of ERK activity for each RAS isoform in response to EGF stimulation.**

A–E ERK activity in each of the 8 MEF cell lines, after growth factor withdrawal for 16–24 h, followed by stimulus consisting of (A) media only, or (B) 10 ng/ml EGF. (A, B) Mean values over 3 replicate cultures. (C) Three example single-cell traces per cell line, randomly selected from 3 replicate experiments. (D) Average baseline (pre-stimulus) ERK activity over 58 replicate cultures per cell line. Each dot represents the median value across cells in an experiment and vertical lines represent the 25th–75th percentiles. Black horizontal bars denote the median across all replicates. Asterisks indicate significance by *t*-test (pFDR < 0.05). (E) Average volatility (pre-stimulus) over 58 replicate cultures per cell line, reflecting the scale of variation over time, displayed as in D. Dots show medians and error bars show 25th–75th percentiles. Asterisks indicate significance by *t*-test (pFDR < 0.05).

F–J Analysis of single-cell response likelihood after EGF stimulus. (F) Demonstration of many cells not responding to EGF stimulus in the Q61R cell line, in a representative experiment. Black lines highlight one responder and one non-responder cell, with 200 individual cell traces shown. (G) Likelihood of single cells responding to EGF stimulus, for each cell line, showing mean of 3 replicates with error bars showing one standard deviation. Asterisks indicate significance by *t*-test (pFDR < 0.05). (H) Relationship between response likelihood and average baseline ERK activity as a possible correlate, for each cell line. Means taken over 3 replicate cultures. (I) Weakness of correlation between baseline ERK activity and response likelihood, measured by Tjur's coefficient of discrimination (i.e., correlation coefficient for a binary response). Inset shows an example from the KRAS<sup>G12D</sup> mutant, where dots are scattered per cell by baseline ERK activity (*x*-axis) and whether that cell responded to EGF (binary *y*-axis). Orange line indicates the logistic fit. Correlations calculated from single-cell data from 3 replicate experiments. (J) Scattered single-cell measurements of baseline ERK activity and amplitude of the change after EGF stimulus. Green triangles: cells that responded; blue circles: cells that did not respond.

K–M Analysis of the response to EGF, by filtering to remove cells that do not respond, presented as in (D), with dots showing medians and error bars showing 25th–75th percentiles. Asterisks indicate significance by *t*-test (pFDR < 0.05). Data from 3 independent culture replicates. (K) Peak ERK activity reached after EGF stimulus. (L) Average ERK activity after 2 h in the presence of EGF. (M) Delay between EGF stimulus and peak ERK activity.

Data information: All *t*-tests herein were performed as detailed in Methods and Protocols, "Statistical Analysis: *t*-tests for Single-Cell Data".

the pathway in the absence or presence of the ERK inhibitor SCH772984 (ERKi; Fig 4A; Morris *et al*, 2013). As an ATP competitive inhibitor with an allosteric mode, ERKi suppresses both the activity of ppERK and the phosphorylation of ERK by MEK (Chaikuad *et al*, 2014). Consistent with this allosteric inhibition, this treatment had a partial effect on ERK phosphorylation (Fig 4A and B). To test for ERK-mediated feedback effects, we treated KRAS[WT], KRAS[G12C], and KRAS[Q61R] cells with 100 nM ERKi 1 h prior to EGF stimulation and measured the phosphorylated or active forms of EGFR, RAS, AKT, and MEK by immunoblotting. In Fig 4B, the comparison between untreated and ERKi-treated cells is shown side by side (dark- and light-shaded bars, respectively) to emphasize the effect of ERK-mediated feedback on each species. Treatment with ERKi resulted in increased MEK dual phosphorylation at Ser217/Ser221 (ppMEK) to different extents under both resting and EGF-stimulated conditions in all three cell lines (Fig 4B), confirming the presence of feedback effects on pathway activity upstream of MEK.

To explore which steps in the pathway are subject to ERK-mediated feedback, we compared immunoblots of ppMEK, EGFR phosphorylation at Tyr1068 (pEGFR), AKT phosphorylation at Ser473 (pAKT), and RAS activation by pulldown of GTP-bound RAS (RAF-RBD PD; Fig 4A and B). In unstimulated mutant and wild-type KRAS cells, only ppMEK was elevated by ERKi treatment, indicating a significant negative feedback effect due to ERK-mediated inhibitory phosphorylation of RAF (Dougherty *et al*, 2005). Upon EGF stimulation of KRAS[WT] cells, we observed the expected increases in pEGFR, pAKT, ppMEK, and RAF-RBD bound RAS. With the exception of pEGFR, all of these species were further increased significantly by ERKi treatment, indicating that under stimulated conditions, ERK-mediated negative feedback also acts at the level of RAS and/or recruitment of GEFs and GAPs, but not receptor activation. These data argue that ERK-mediated negative feedback is distributed throughout the pathway to constrain ERK activation.

However, a different pattern was observed in stimulated mutant KRAS[G12C] and KRAS[Q61R] cells. While pEGFR, pAKT, and ppMEK were all significantly increased by EGF stimulation, no significant increase was detected in RAF-RBD pulldown of RAS for either mutant (Fig 4A and B). The increases in pEGFR and ppMEK, though significant, were lower in magnitude in both KRAS[G12C] and KRAS[Q61R] compared with KRAS[WT], indicating a decreased pathway responsiveness relative to baseline, which is consistent with reduced ERK activity responses for these cell lines (Fig 3). The EGF-stimulated increase in ppERK in KRAS[G12C] cells was similar to KRAS[WT] cells, but reduced in KRAS[Q61R] cells, also consistent with EGF-stimulated ERK activity measurements for these cell lines. As in the KRAS[WT] cells, negative feedback was assessed by the increase in EGF-stimulated phosphorylation of each protein in the presence of ERKi relative to the vehicle treatment. Unlike KRAS[WT] cells, RAF-RBD-bound RAS was not further increased by ERKi. Similarly, ppMEK and pAKT were increased by ERKi treatment to a much lesser degree in KRAS[G12C] and KRAS[Q61R] cells than in KRAS[WT]. These data suggest that ERK-mediated feedback is weaker in mutant cells relative to wild type, although they do not rule out the possibility that in the mutant cells, one or more steps in the pathway reaches saturation under these conditions, limiting the ERKi-driven increase.

## ERK activity is rescaled bidirectionally, independent of pathway expression levels

The similarity in ERK activity between wild type and mutant KRAS cells contrasts starkly with the conceptual model of RAS mutations hyperactivating the pathway. As this difference is not clearly explained by direct feedback effects from ERK, we employed a mathematical modeling approach to more carefully consider other variables that could account for it (Fig 5A). In a simple linear view of RAS-to-ERK transduction, the mutant RAS GTPase activities being 50- to 800-fold lower than wild type would be expected to produce correspondingly large changes in both pre-stimulus and stimulated ERK activity. However, additional variables may modulate the mutant activity. First, variation in expression level of pathway components could compensate for the differences in RAS activity, especially if expression levels become limiting. Second, the affinities of GTP-bound RAS mutants for their effectors are not equivalent to wild type; mutation lowers the affinity for RAF up to 7-fold (Hunter *et al*, 2015). We therefore designed our modeling approach to predict the expected increase in ERK activity based on known properties of KRAS mutants.

Our modeling approach requires the relative protein concentration for components of the pathway, which we evaluated using immunoblots. Initial experiments indicated that the cell lines varied both in the total amount of protein extracted by our protocol, and in the abundance of typical loading control proteins such as actin. We therefore employed an alternative approach to normalize each sample by total protein (see Materials and Methods). Using this approach, each cell line was assessed at baseline, peak (~ 15 min), and steady state (~ 2 h) following a 10 ng/ml EGF stimulus (Fig 5B). In these samples, we also measured the fraction of ERK that is dually-phosphorylated via Phos-Tag immunoblot (Aoki *et al*, 2013). This measurement of ppERK confirmed the trends observed by FRET measurements: EGF induced a peak in ppERK which then diminished at steady state, and ppERK was elevated in mutant RAS cells only under baseline conditions (Fig 5C and D). As expected, measurements of ppERK were well correlated with the fraction of dually-phosphorylated ERK (Fig EV1). This quantitative dataset rules out two simple explanations for low ERK activity in KRAS mutants. First, compared with KRAS[WT] cells, total RAS levels were higher in KRAS[G12C] and KRAS[G12V] lines, precluding the possibility that the expression level of RAS was compensating for the excess mutant activity (Fig 5B). Second, levels of BRAF, MEK, and ERK, but not CRAF, varied significantly among cell lines, but without a clear pattern or correlation structure. When total expression levels were compared with ERK activity (via EKAR), no significant correlations were found, suggesting that no individual component acts as a limiting factor. Moreover, dually-phosphorylated ERK ranged from only ~ 1 to ~ 30% of total ERK across all samples (Fig 5C), confirming that it does not reach saturation.

We parameterized our model using the measured protein levels, along with the biochemical activity of wild type and mutant KRAS proteins previously reported *in vitro* (Gremer *et al*, 2011; Smith *et al*, 2013; Hunter *et al*, 2015; Appendix Table S1). As the goal of our modeling approach was to identify potential explanations for restrained ERK signaling in KRAS mutant cells, we omitted any additional regulation, such as feedback. Effectively, the model separates operation of the core GTPases and kinases of the cascade,

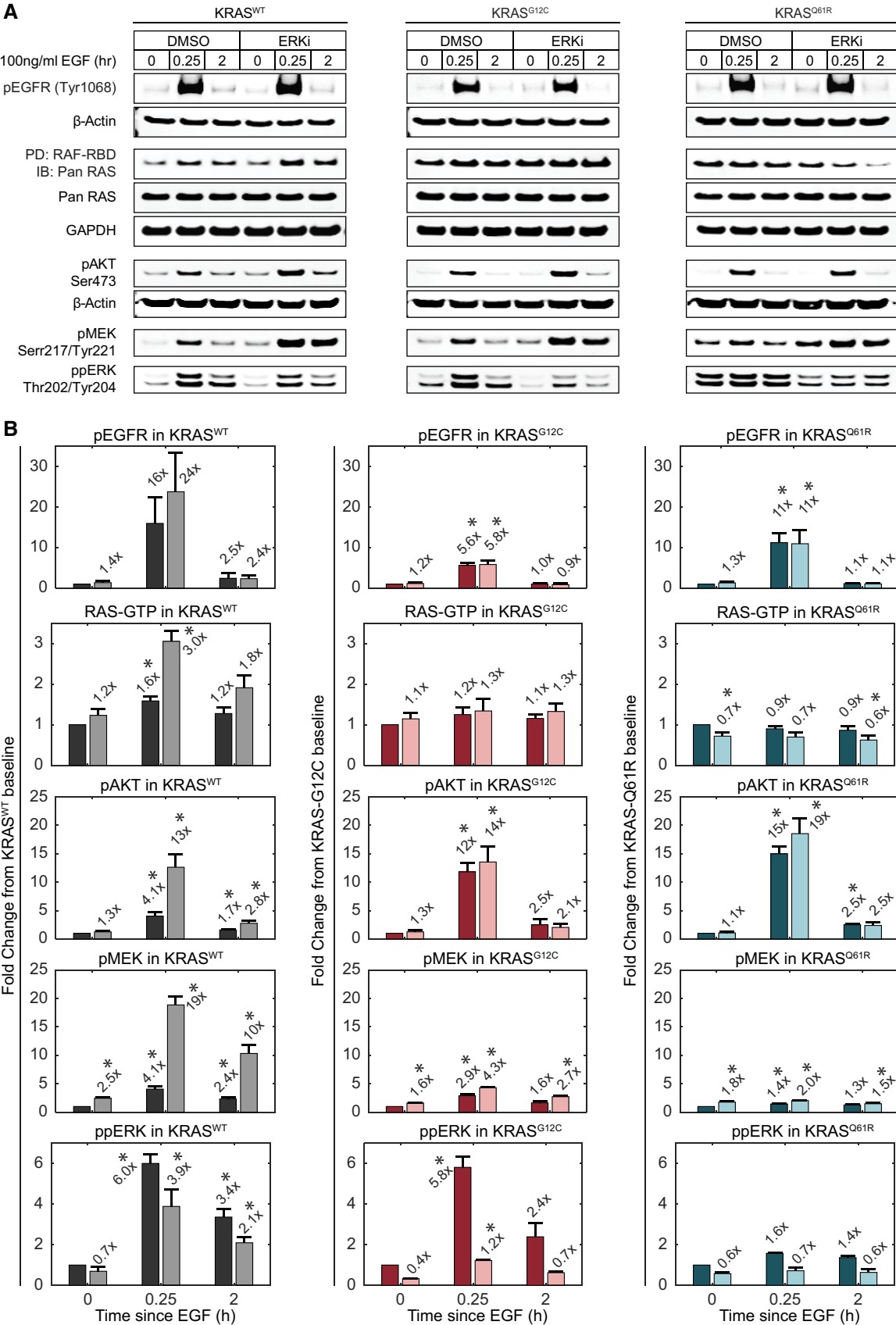

**Figure 4.**

**Figure 4. Analysis of ERK-dependent feedback in RAS mutants.**

A, B Immunoblot analysis of RAS-ERK pathway activity at multiple levels in the absence or presence of ERK inhibition by 100 nM SCH772984 (ERKi). Lysates for the indicated cell lines were collected at baseline, peak (15 min), and steady state (2 h) time points after treatment with 100 ng/ml EGF. (A) Sample blot imagery for each measurement. (B) Quantified measurements, shown as fold change relative to the DMSO-treated baseline sample. Values for pEGFR, pAKT, ppMEK, and ppERK were normalized to β-actin; data for RAF-RBD PD/Pan RAS were normalized to total Pan RAS. Bars represent the mean of triplicate measurements and error bars the standard error of the mean. Mean fold change values are printed above each bar. Dark bars: DMSO-treated; light bars: ERKi-treated. *x*-axis indicates the duration of EGF treatment. Asterisks indicate statistical significance by *t*-test (*P* < 0.05).

Source data are available online for this figure.

which we term "internal" factors, from "external" factors that that include feedbacks, adapters, scaffolding proteins, and/or phosphatases. By comparing model predictions against experimental data, this approach identifies the differences attributable to external factors. Using a steady-state solution of this model, we predicted the baseline and EGF-stimulated steady-state levels of ppERK for the isoforms for which biochemical data is available: $KRAS^{WT}$, $KRAS^{G12C}$, $KRAS^{G12D}$, $KRAS^{G12V}$, and $KRAS^{Q61R}$ (Fig 5E). These simulations confirmed that the experimentally measured ppERK is indeed much lower in KRAS mutants than expected, especially at baseline. Conversely, the amplitude (fold change) in ppERK upon stimulation is greater in the experimental system than in the model, except for $KRAS^{Q61R}$ where differences are indistinguishable (Fig 5F). This analysis clarifies the role of the external factors in the ERK pathway, revealing that they have a bidirectional effect: They suppress ppERK under both baseline and stimulated conditions, but also amplify the difference between these conditions. The external factors therefore effectively increase the responsiveness of ERK to growth factor stimulation.

To gain further insight into the relative importance of internal and external factors modulating ERK activity, we extended our analysis to test whether ppERK correlates with expression level of pathway components, using partial least squares regression (PLSR). We fit both the simulated and measured ppERK against the measured protein expression and the presence of EGF stimulation. PLSR explained 75% of the variance in simulated ppERK, but only 55% of the variance in experimentally measured ppERK (Fig 5G). In the simulated ppERK data, we found significant correlations with the abundance of RAS, MEK, and ERK proteins. In contrast, ppERK in the experimental system was only significantly correlated with the presence of growth factor stimulation (Fig 5H). Thus, another key function of external factors is to confer robustness to expression level variation in cascade components, extending previous observations that the regulation of ERK phosphorylation is robust to changes in ERK expression level (Fritsche-Guenther *et al*, 2011). Altogether, our model analysis reveals that external factors increase the dynamic range of ERK response to EGF and that nearly all of this control lies in mechanisms outside of the linear RAS-to-ERK kinase cascade. This bidirectional effect on ERK activity and the moderate strength of feedback observed in mutant KRAS cells (Fig 4) imply that ERK activity in the cell is rescaled by mechanisms beyond simple negative feedback (Dougherty *et al*, 2005; Amit *et al*, 2007).

**Phosphatases dynamically shape the functional ERK output**

Multiple phosphatases are dynamically regulated during growth factor responses (Amit *et al*, 2007), some directly by ERK (Yoon & Seger, 2006), raising the question of whether such regulation could

contribute to the observed rescaling of ERK dynamic range. While phosphatase protein levels can be quantified, phosphatase activity is typically difficult to assay, especially in living cells. However, our experimental system provided a unique opportunity to estimate phosphatase activity acting on ERK substrates by comparing the datasets for ERK activity (measured by FRET, Figs 2 and 3) and the abundance of active ERK molecules (measured by immunoblot, Fig 4). While these measures are typically considered equivalent under the assumption that phosphatase activity should be stable, our measurements made under identical conditions reveal some differences. For example, we observe significant variation in ppERK across cell lines, while ERK activity is indistinguishable in the same lines, especially at the steady-state time point (comparing Figs 3K and L to 5D). As noted in our calibration of EKAR, the activity measurement is a ratio of the concentrations of active ERK and any phosphatases that act on ERK substrates. We therefore inferred how this phosphatase activity varied by examining the correspondence between ppERK and activity measurements (Fig 6A) and estimating the relative phosphatase activity as the ratio of these values (Fig 6B). All data used for these ratios fell within the linear range for the respective measurements (see Figs 1E and EV1).

At baseline, phosphatase activity appears nearly uniform across cell lines, except for the $KRAS^{WT}$ measurement, which is likely an outlier (Fig 6B). This consistency in phosphatase activity between cell lines results in a significant linear correlation between ppERK and ERK activity, as expected (Fig 6A). However, after stimulation, both the slope and correlation are diminished, and variance in the estimated phosphatase activity increases. By the steady-state time point, estimated phosphatase activity rises significantly, as the different RAS cell lines settle to very similar levels of ERK activity (Fig 3L) despite varying levels of ppERK (Fig 5D). This correlational analysis implies that after stimulation, phosphatase activities and/or levels are regulated in such a manner that they act to normalize the levels of ERK activity, despite residual differences in concentration of ppERK. However, the observation that apparent phosphatase activity at steady state is uncorrelated to ppERK at baseline implies dynamic complexity beyond simple regulation by ERK.

To corroborate the implied regulation of phosphatase activity, we used an independent indicator of phosphatase activity, the decay of EKAR FRET signal following MEK inhibition. Because all of our experiments included MEK inhibitor as a final treatment, we were able to fit decay curves for individual cells with single exponential functions, whose time constant is in principle proportional to the phosphatase activity acting on the reporter (Fig 6C). Across all cell lines and treatments, a pattern emerges of increased phosphatase activity in KRAS mutants and increases, with some treatments, in cells expressing a wild-type RAS (Fig 6D), though the current dataset lacks sufficient replication to establish statistical significance

across the many conditions. We performed a more focused statistical analysis of the decay rates, comparing each cell line to KRAS[WT] at baseline (Fig 6E), and comparing each treatment to the baseline within KRAS[WT] (Fig 6F). This analysis demonstrates significantly elevated activity at baseline in several mutant cell lines, and with FGF treatment in KRAS[WT], supporting the indication that phosphatases are differentially regulated. While these analyses are

specific to phosphatases acting at the level of ERK substrates, it is likely that a similar mechanism is functionally relevant at the level of ERK or MEK, contributing to the bidirectional modulation of ERK activity in cell lines with severe KRAS mutations. We validated this concept by immunoblot analysis examining DUSP6, a phosphatase that acts on ERK1/2 (Fig 6G). The level of DUSP6 is consistently elevated in the presence of ERK inhibitor, compared to baseline

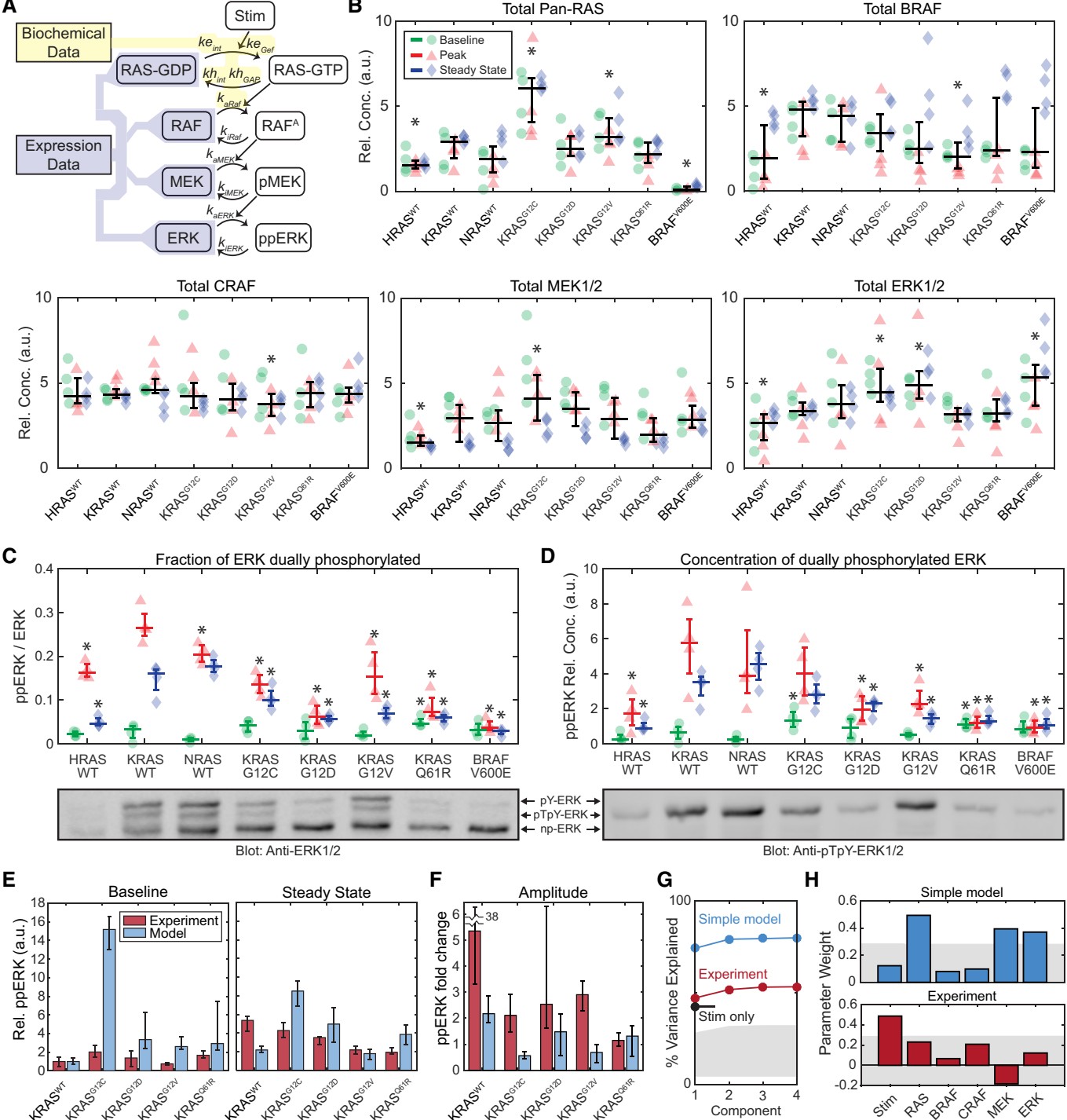

**Figure 5.**

◀

**Figure 5. Quantitative analysis of ERK phosphorylation in response to RAS mutation.**

A Schematic of a model of the internal factors of the RAS-ERK pathway, showing parameters associated with each reaction. Shaded regions indicate portions of the model for which parameter values are available from either (yellow) published biochemical assays or (blue) our immunoblot expression data (B–D).

B Immunoblot measurement of RAS-ERK pathway components in each cell line, at baseline (green circles), peak activity (15 min, red triangles), and steady-state activity (2 h, blue diamonds), four replicates each. Overlaid plots indicate the median (horizontal bars) and $25^{th}$–$75^{th}$ percentiles (vertical whiskers) over all conditions. Asterisks indicate statistical significance from the KRAS$^{WT}$ cell line, by $t$-test (pFDR < 0.05).

C, D Phos-Tag immunoblot measurement of ERK fractional phosphorylation (C) and the relative concentration of dually-phosphorylated ERK (D), annotated as in (B), but with median and percentile ranges indicated per treatment condition, for 4 replicate cultures. Sample blot imagery shows anti-ERK1/2 (below C) and anti-ppERK1/2 (below D) for the same blot replicate. Asterisks indicate significance by $t$-test (pFDR < 0.05) from the KRAS$^{WT}$ cell line measurement for each condition.

E, F Comparison of the internal RAS-ERK model to experimental data. (E) Relative ppERK as predicted by the internal model and measured by immunoblot for the 4 replicates collected, showing the median of the baseline and steady state after EGF treatment (E), and the amplitude of stimulation (F). Error bars show $25^{th}$–$75^{th}$ percentiles.

G, H Partial least squares regression of both experimental ppERK measurements and predictions via the simple RAS-ERK model. Regression was based on presence/absence of EGF, and expression levels of RAS, BRAF, CRAF, MEK, and ERK. (G) Percent of variance explained by each PLS model considered, based on how many component terms are allowed. Stim only refers to a PLS model using experimental ppERK data, but only predicting based on the presence/absence of EGF. (H) Weights assigned to each parameter in the PLS models. Gray shaded regions indicate the bounds of statistical significance, determined via bootstrapping with scrambled data. Only values that extend beyond the gray regions are statistically significant from zero ($P$ < 0.05).

Source data are available online for this figure.

without growth factor. Furthermore, DUSP6 levels are rapidly and transiently suppressed on stimulation with EGF in the KRAS$^{WT}$ and KRAS$^{G12C}$ lines, an effect which appears at least partially mitigated by ERK inhibition. While DUSP6 is known to be primarily specific for ERK due to a direct binding interaction, its dynamic regulation demonstrates that phosphatase expression can shift on a time scale consistent with the inferred phosphatase activity. Additional work will be needed to more clearly identify the phosphatases that act on EKAR or on endogenous ERK substrates.

## Discussion

Here, we used a single-cell approach to bring increased temporal resolution and quantitative rigor to the question of how oncogenic RAS mutants alter signaling behavior within the cell. Our analysis provides two major conclusions. First, our systematic dataset reconciles previously conflicting observations of ERK activity driven by RAS mutations. Second, we find that mutant-driven ERK activity is not simply suppressed by negative feedback as previously postulated, but instead that multiple mechanisms cooperate to constrain its dynamic range in response to stimuli.

### A unified model of ERK activity as stimulated by growth factors and mutant RAS activity

The canonical view is that RAS mutations hyperactivate the ERK/MAPK cascade. Yet, several experimental models of the conversion of a single *Kras* allele from wild type to GTPase-defective mutant have found that this alteration results in no increase, or even a decrease, in activated ERK (Guerra *et al*, 2003; Tuveson *et al*, 2004; Konishi *et al*, 2007; Huang *et al*, 2014). While similar ERK signaling could result from negative feedback that restrains mutant-driven activity, it is less clear what mechanisms would result in lower ERK activity.

Based on the data presented here, these differences can now be attributed to the temporal and quantitative limitations of the methods previously used to measure ERK activation (primarily uncalibrated ppERK immunoblots). Our dataset recapitulates the reported attributes of mutant KRAS signaling that in isolation appear

contradictory: elevated baseline signaling, retained capacity for GF stimulation, and reduced absolute peak upon stimulus. Additionally, we find that RAS mutant cells have a reduced probability of response that is independent of their current ERK activity, especially in the case of the severe Q61R mutation. This tendency toward unresponsiveness may contribute to the reduced ERK activation observed in RAS mutant cells in the presence of serum or growth factors.

In addition to reconciling previous observations, our analysis also reveals a previously unquantified phenomenon, which is that ERK activity remains unexpectedly responsive to growth factor stimulation in cells carrying mutant KRAS (Fig 7). This responsiveness arises because, while both baseline and peak GF-stimulated ERK activity are limited in KRAS mutant cells, the relative strength of suppression is greater at baseline than at peak. This observation is consistent with the idea that dynamic range of a signaling pathway is a physiologically important parameter (Janes *et al*, 2008) and that mechanisms exist to buffer it from deleterious mutations. This effect is only reliably observable through comparisons between isogenic mutant and non-mutant cells using calibrated ERK activation measurements, underscoring the importance of a quantitative, systematic approach to complex signaling networks.

### The MAPK pathway as a robust interpreter moderating mutant RAS signaling

The consistency of ERK signaling in the context of RAS mutations or changes in pathway expression has been attributed mainly to negative feedback from ERK (Courtois-Cox *et al*, 2006; Fritsche-Guenther *et al*, 2011). However, our systematic analysis reveals a more complex situation with differential suppression of ERK distributed across multiple factors. ERK-mediated negative feedback plays a significant role in restraining MEK activation, but that role appears lesser in mutant KRAS cells rather than greater (Fig 4). At least two models may explain the bidirectional rescaling of ERK signaling that we observe. One possibility is that the RAF/MEK/ERK cascade could act as fold change detector for variation in EGFR activity (Cohen-Saidon *et al*, 2009). Fold change detector models are expected to incorporate motifs such as an incoherent feedforward loop or a nonlinear integral feedback loop (Adler *et al*, 2017). An alternative

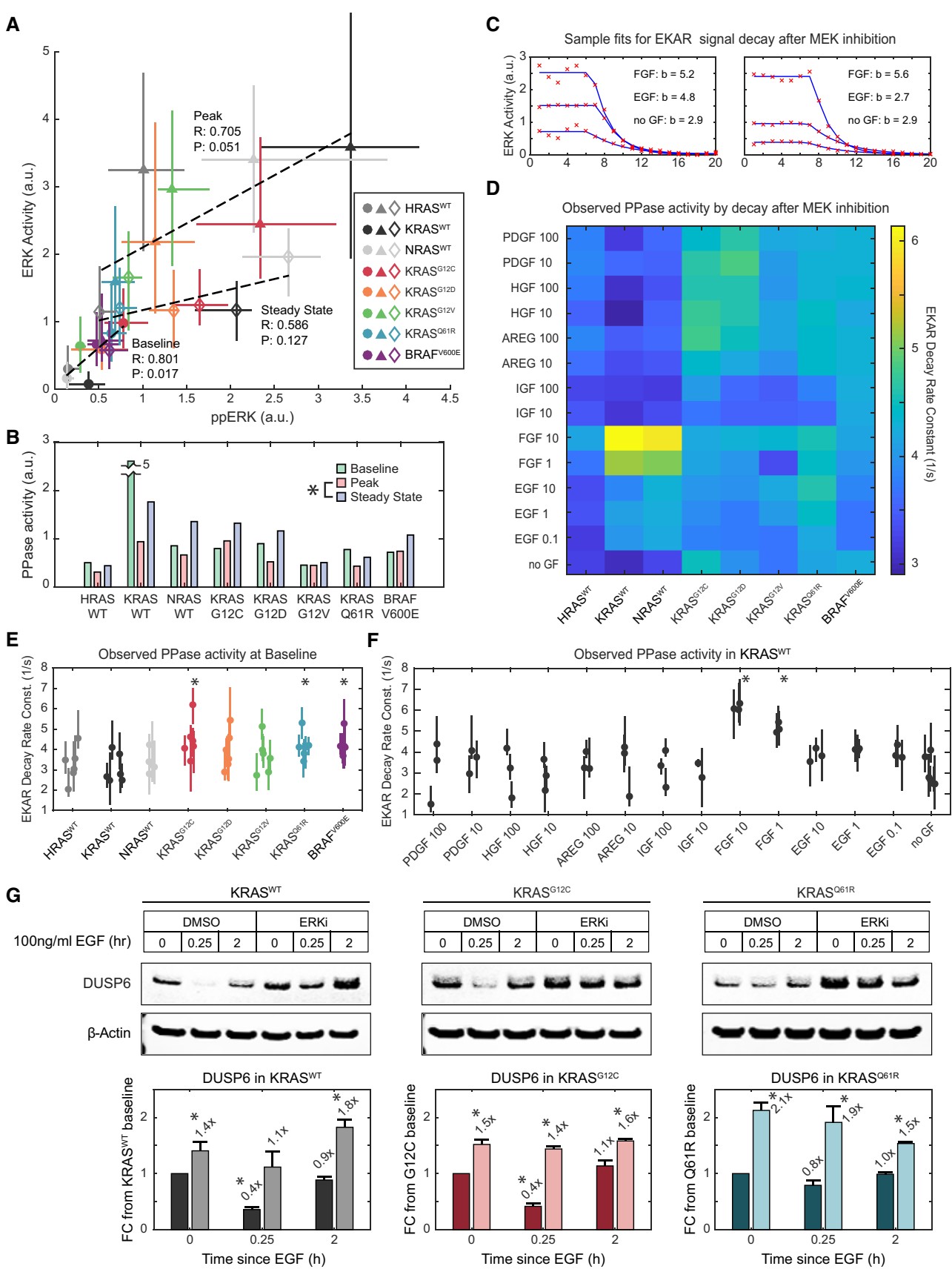

**Figure 6.**

**Figure 6. Inference of phosphatase activity on ERK substrates.**

A    Correlation of ERK activity and ppERK concentration, median of 3 and 4 replicates, respectively, per condition. Error bars denote $25^{th}$–$75^{th}$ percentiles, including single-cell distributions for ERK activity. Markers are color-coded by cell line, and marker shape indicates treatment (circle: baseline, triangle: peak, diamond: steady state). Dotted lines show linear regression for each treatment; Pearson's correlation coefficients ($R$), and associated $P$-values ($P$) are printed alongside.

B    Estimate of substrate level phosphatase activity per cell line and treatment, calculated as ppERK/ERK activity. Asterisk indicates significance when comparing all cell lines, by $t$-test (pFDR < 0.05).

C    EKAR signal decay after MEK inhibition, example single-cell data (red x's) fit to a decaying exponential model (blue lines), with decay rate constants (b) printed.

D    Heatmap of median decay rate constants fit for each cell line and treatment.

E, F  Statistical analysis of phosphatase activities observed by EKAR signal decay at (E) baseline (i.e., no GF treatment prior to MEKi) for all cell lines compared with KRAS^WT, with 6 replicates, and (F) in KRAS^WT for all treatments, compared with no GF, with 3 replicates. Dots denote median values, and bars $25^{th}$–$75^{th}$ percentiles. Asterisks indicate significance by $t$-test (pFDR < 0.05); $t$-tests performed as detailed in Methods and Protocols "Statistical Analysis: $t$-tests for Single-Cell Data".

G    Immunoblot analysis of DUSP6 levels, subject to stimulation by EGF and inhibition of ERK by 100 nM SCH772984. Bars represent the mean of triplicate measurements and error bars the standard error of the mean. Asterisks indicate statistical significance by $t$-test ($P$ < 0.05).

Source data are available online for this figure.

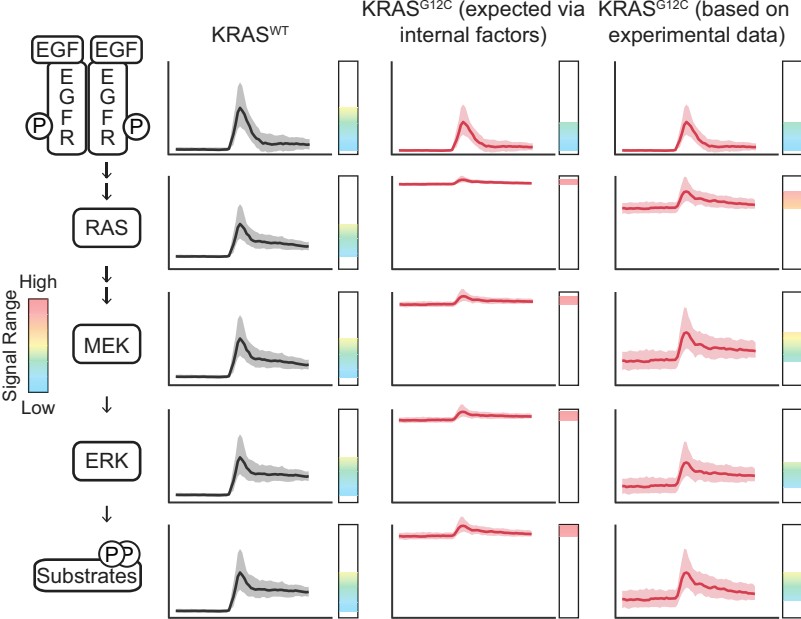

**Figure 7. Rescaling and expansion of dynamic range through the RAS/ERK pathway.**

Diagram depicts the activities we expect at various levels in the RAS/ERK pathway, as time series in our typical EGF stimulation experiment, extrapolated from our live-cell data and immunoblot measurements. From left to right, columns depict the response in KRAS^WT, the expected response in KRAS^G12C cells (as an example KRAS mutant) considering only the internal factors of the RAS-ERK pathway, and observed behavior of KRAS^G12C cells. Vertical bars indicate the dynamic range of the activity of that node in the network, with the colored spectrum indicating signal levels from low (blue) to high (yellow-orange) and excessive (red).

model is that the pathway is arranged for dose-response alignment (Brent, 2009), which ideally employs push–pull mechanisms or combines negative feedback with a comparator (Andrews *et al*, 2016). In both models, ERK-mediated negative feedback could play a critical part but would operate in concert with additional regulatory interactions. Such models could explain the complex behavior we observe of restraint of excessive activity from mutant KRAS that also preserves and enhances the ability of the pathway to respond to growth factor stimulus.

Another factor affecting the output of the pathway is the apparent fine-tuning of phosphatase activity acting on ERK substrates. As ERK phosphorylation is often used as a *de facto* measurement of its activity, quantitative effects at the level of substrates have received less attention. Nonetheless, the ability of ERK to maintain phosphorylation of its substrates is inherently limited by the opposing process of dephosphorylation, making this a critical but understudied control point. Our data imply that regulation of this process is significant for an exogenous FRET-based substrate whose sequence is based on the endogenous substrate Cdc25A, warranting further study of this effect on endogenous substrates. This effect could be mediated by direct control of phosphatase activity, or through competition of substrates for the phosphatase (Rowland *et al*, 2015); future work will be needed to elucidate this mechanism.

Lastly, the potential for each RAS variant line to have been subject to selection during the process of cell line construction and propagation may play a significant role (Li *et al*, 2018). Cells receiving a RAS insertion that produces sufficiently high levels of expression to drive truly excessive ERK activity could be driven into senescence and thus prevented from establishing a cell line. Therefore, cells bearing epigenetic modifications or point mutations that

moderate the output of ERK could be overrepresented in the surviving population. While our analysis indicates that the expression level of pathway components is insignificant in determining ERK activity in the cell lines assayed, this rules out neither activity-modifying mutations that do not alter expression, nor the existence of an activity threshold above which cells are eliminated by selection. Furthermore, while our limited immunoblot-based analysis was unable to identify differences in expression that explain the divergence of ERK signaling from the expected, a more precise and comprehensive proteomic analysis could reveal overlooked correlations (Shi et al, 2016). Naturally, the same caveats apply to the vast majority of cell-based experiments on RAS signaling (including transient expression experiments that typically exceed at least one cell cycle). Thus, experimental strategies in which RAS isoforms are abruptly exchanged, and the resulting cellular changes monitored with high temporal resolution, would be informative in understanding the adaptation to a RAS mutation.

### Constraints on RAS-driven signaling in oncogenesis

The ability of the ERK/MAPK pathway to constrain the quantitative effects of mutant KRAS raises important questions for how these mutations function in oncogenesis. In many cancers, RAS mutations are thought to occur very early in oncogenesis, and therefore, the homeostatic nature of the pathway likely plays a central role in determining whether a RAS mutant cell progresses toward malignancy (Li et al, 2018). Our data from cells with few other genetic abnormalities can be a considered a model for signaling at this early stage, unique from studies that have investigated mutant RAS in fully developed cancers and focused on treatment of later-stage disease. However, if mutations in RAS lead to only modest changes in ERK signaling, how do they drive progression toward malignancy? One potential model was that excess ERK activity could engage lower-affinity substrates, expanding the effective ERK-driven phosphoproteome to non-traditional targets. However, given the constraints we observe on the magnitude of ERK signaling, it is impractical for these KRAS mutant cells to promote phosphorylation of non-typical ERK targets. Furthermore, KRAS mutant-bearing cells do not show longer duration of peak ERK activity following stimulus

than those with $KRAS^{WT}$, so excess activation after growth factor stimulus is also unlikely. Instead, our finding that the over-activating effect of KRAS mutants is limited to chronic baseline elevation implies (i) that chronic moderate signaling is sufficient to drive deleterious phenotypes and (ii) that mutant cells are unlikely to respond to normal low-level signaling.

A strong downstream effect from chronic moderate ERK activity is consistent with current models of some effectors. The ERK target gene Fra-1, a transcription factor whose expression is correlated with cancer invasiveness (Tam et al, 2013), integrates ERK activity over time (Gillies et al 2017). With its slow decay rate [half-life > 5 h (Basbous et al, 2007)], Fra-1 can accumulate to relatively high levels over a long period of moderately elevated ERK activity. Any ERK-induced gene products with similar degradation kinetics will also accumulate over time in cells with baseline ERK elevation. Conversely, gene products subject to rapid degradation kinetics such as c-Fos and Egr-1 would be only weakly elevated in RAS mutants, compared to the large changes in expression driven by sporadic wild-type activity. Products under negative regulation, such as those that degrade rapidly even with extended activity (Wilson et al, 2017), may actually be suppressed by chronic ERK activity. Thus, while enhanced ERK kinase activity as an indicator of early RAS mutant cells is difficult to detect without live-cell measurements, the resulting expression profile—particularly the ratio between long-term and short-term responsive genes—may be more informative.

While the damping of mutant RAS-driven signals at the level of ERK may appear to be a tumor suppressive mechanism; this is not necessarily the case. KRAS mutation frequencies in human cancer and data from mouse models suggest that a limited quantitative range of RAS signal (a "sweet spot") is critical for the development of tumors (Sarkisian et al, 2007; Li et al, 2018). Pathway constraints could help RAS mutant cells to stay within this range and evade senescence or cell death due to excessive ERK activation. This paradox raises the question of whether RAS mutations are more common than downstream mutations (such as MEK or ERK) in cancer and related syndromes such as RASopathies because they are strong enough to induce increased ERK activity, or rather because they are more constrained and able to escape selection by senescence.

# Materials and Methods

### Reagents and Tools table

| Reagent or resource | Source | Identifier |
|---|---|---|
| **Antibodies** | | |
| Anti-ERK1/2, Rabbit | CST | 9102 |
| Anti-ERK1/2, Mouse | CST | 4696 |
| Anti-phospho-ERK1/2(Y202/Y204), Rabbit | CST | 4370 |
| Anti-MEK1/2, Mouse | CST | 4694 |
| Anti-phospho-MEK1/2(S217/S221), Rabbit | CST | 9121 |
| Anti-CRAF, Mouse | CST | 12552 |
| Anti-BRAF, Mouse | Invitrogen | MA5-15495 |
| Anit-panRAS, Mouse | Cytoskeleton | AESA02 |
| Anti-phospho-AKT(S473) | CST | 4060 |

**Reagents and Tools table** (continued)

| Reagent or resource | Source | Identifier |
|---|---|---|
| Anti-beta-Actin | Sigma-Aldrich | A2228 |
| Anti-Tubulin | CST | 3873 |
| Anti-GAPDH | CST | 2118 |
| IRDye 800CW Donkey anti-Rabbit IgG | Licor | 925-32213 |
| IRDye 800CW Donkey anti-Mouse IgG | Licor | 926-32212 |
| IRDye 680RD Donkey anti-Rabbit IgG | Licor | 926-68073 |
| IRDye 680RD Donkey anti-Mouse IgG | Licor | 925-68072 |
| **Chemicals, peptides, and recombinant proteins** | | |
| Amphiregulin | Peprotech | 100-55B |
| EGF | Peprotech | AF-100-15 |
| FGF | Peprotech | 100-18B |
| HGF | Peprotech | 100-39 |
| IGF-I | Peprotech | 100-11 |
| PDGF-AB | Peprotech | 100-00AB |
| ARS-853 | MedChemExpress | HY-19706 |
| PD0325901 | Selleck Biochemicals | S1036 |
| SCH772984 | Selleck Biochemicals | S7101 |
| Blasticidin | Corning | 30-100-R1 |
| Bovine serum albumin | Sigma-Aldrich | A7906 |
| Collagen I, rat tail | Life Technologies | A10483-01 |
| Dextrose | Fisher | C6H1206 |
| L-Glutamine | Life Technologies | 25030-081 |
| Puromycin dihydrochloride | Sigma-Aldrich | P8833 |
| DMEM | Life Technologies | 11965-092 |
| Fetal bovine serum | Gemini Bio-products | 100-106 |
| Penicillin streptomycin | Life Technologies | 15070-063 |
| 0.25% Trypsin-EDTA | Life Technologies | 25200-056 |
| Tris Base | Fisher | BP152 |
| Glycine (Crystalline Powder) | Fisher | BP381 |
| Sodium dodecyl sulfate (SDS), Micropellets | Fisher | BP8200500 |
| Ponceau S solution, suitable for electrophoresis, 0.1% ($w/v$) in 5% acetic acid, 1L | Sigma-Aldrich | P7170-1L |
| Bromophenol blue | Sigma-Aldrich | B5525 |
| Dithiothreitol | Fisher | BP172 |
| **Critical commercial assays** | | |
| Amaxa MEF 2 Nucleofector kit | Lonza | VPD-1005 |
| **Experimental models: cell lines** | | |
| RAS-less MEF (Mouse Embryonic Fibroblast) cell lines with transduced human RAS or BRAFV600E | Gift from Dom Esposito, Frederick National Laboratory | |
| HRAS[WT] EKAR3 | This paper | Available on request |
| KRAS4B[WT] EKAR3 | This paper | Available on request |
| NRAS[WT] EKAR3 | This paper | Available on request |
| KRAS[G12C] EKAR3 | This paper | Available on request |
| KRAS[G12D] EKAR3 | This paper | Available on request |

**Reagents and Tools table** (continued)

| Reagent or resource | Source | Identifier |
|---|---|---|
| KRAS$^{G12V}$ EKAR3 | This paper | Available on request |
| KRAS$^{Q61R}$ EKAR3 | This paper | Available on request |
| BRAF$^{V600E}$ EKAR3 | This paper | Available on request |
| **Recombinant DNA** | | |
| Plasmid: pPBJ-EKAR3nls-puro | Sparta *et al* (2015) | Addgene # forthcoming |
| **Software and Algorithms** | | |
| NIS Elements AR ver. 4.20 | Nikon | RRID:SCR_014329 |
| Bio-Formats ver. 5.1.1 (May 2015) | OME | RRID:SCR_000450 |
| uTrack 2.0 | Jaqaman *et al* (2008) | http://www.utsouthwestern.edu/labs/danuser/software/ |
| MATLAB | Mathworks | SCR_001622 |
| **Other** | | |
| Glass-Bottom Plates, #1.5 cover glass | Cellvis | P24-1.5H-N, P96-1.5H-N |
| 12% Mini-PROTEAN® TGX™ Precast Protein Gels, 15-well, 15 μl | Bio-Rad | 4561046 |
| SuperSep Phos-tag gels (50 μmol/l), 12.5%, 17 wells | Wako-Chem | 195-17991 |
| GE Healthcare Amersham™ Protran™ NC Nitrocellulose Membranes: Rolls, 0.1 μm pore | Fisher | 45-004-000 |

## Methods and Protocols

### Cell culture

Mouse embryonic fibroblasts expressing a single RAS isoform were obtained from the Frederick National Laboratory of the National Cancer Institute, Frederick, MD. Cells were authenticated through Whole Exome Sequencing, PCR, and immuno blot methods at the Frederick National Laboratory. Mycoplasma testing was performed on a regular basis with negative results of no contamination. Cells were cultured in DMEM supplemented with 0.2% bovine serum albumin (BSA) and 2.5 μg/ml puromycin or 4 μg/ml blasticidin. For imaging experiments, cells were cultured in a custom imaging media composed of DMEM lacking phenol red, folate and ribo-flavin, glucose, glutamine, and pyruvate, supplemented with 0.1% BSA, 4 mM L-glutamine, and 25 mM glucose.

### Reporter cell line construction

Cells were electroporated using a Lonza Nucleofector electroporator. EKAR3 was stably integrated into cells using the piggyBAC transposase system (Pargett *et al*, 2017). Positive integrants were selected by fluorescence-based cell sorting.

### Live-cell microscopy

Multi-well plates with #1.5 glass bottoms were coated with collagen and seeded with reporter cell lines 1 day prior to imaging. Prepared culture plates were imaged on a Nikon Ti-E inverted microscope with a stage-top incubator to maintain the culture at 37°C and 5% CO$_2$ throughout the experiment. Microscopy and image processing performed as described in (Pargett *et al*, 2017). Imaging sites within each well were selected and imaged sequentially at each acquisition time, automated via the NIS Elements AR software. Images were captured using a 20×/0.75 NA objective and an Andor Zyla 5.5 scMOS

camera. Filter sets used were Chroma #49001 (ET-CFP) and #49003 (ET-YFP).

### Immunofluorescence microscopy

After growth and treatment as indicated on glass-bottom 96-well plates, cells were fixed for 30 min at room temperature with a freshly prepared solution of 12% paraformaldehyde in PBS and permeabilized with 1% Triton X-100. Samples were then stained with primary and secondary antibodies in PBS + 0.1% Triton X-100 + 2% bovine serum albumin, and images were captured on a Nikon Ti-E inverted microscope with a 20×/0.75 NA objective with an Andor Zyla 5.5 scMOS camera.

### Image processing

Imaging data were processed to segment and average pixels within each identified cell's nucleus and cytoplasm, using a custom procedure written for MATLAB (Pargett *et al*, 2017). The procedure accessed image data from ND2 files generated by NIS Elements, using the Bio-Formats MATLAB toolbox, and tracked single-cell positions over time using uTrack 2.0 (Jaqaman *et al*, 2008). The resulting single-cell time series traces were filtered for quality (minimum length of trace, maximum number of contiguous missing, or corrupt data points), and ratiometric reporter levels calculated. EKAR3 level was calculated as $1 - ((CFP/YFP)/R_P)$ as, where $CFP$ and $YFP$ are the pixel intensities of the cyan and yellow channels, respectively, and $R_P$ is the ratio of total power collected in cyan over that of yellow (each computed as the spectral products of relative excitation intensity, exposure time, molar extinction coefficient, quantum yield, light source spectrum, filter transmissivities, and fluorophore absorption and emission spectra). See Appendix Supplementary Methods for detailed interpretation of the EKAR3 signal.

 

### Immunoblotting

For immunoblot experiments, assaying pathway activity and feedback sensitivity (all blots in Fig 4), cells were seeded at a density of $2.5 \times 10^6$ cells per 10 cm plate and starved of growth factor for 6 h in imaging media. Cells were pre-treated with DMSO or 100 nM SCH772983 (Selleckchem) (Morris *et al*, 2013) for the last hour of starvation. Cells were then stimulated with vehicle or 100 ng/ml EGF for 15 min or 2 h and lysed with Cell Lysis Buffer (50 mM Tris pH 7.5, 10 mM $MgCl_2$, 0.5 M NaCl, 2% Igepal) (Cytoskeleton Inc.) containing protease and phosphatase inhibitors (Pierce/Thermo Fisher Scientific). Lysates were clarified by centrifugation at $> 5,000\,g$ for 2 min at 4°C and snap-frozen in liquid nitrogen with protein concentrations measured using the BCA protein assay (Pierce/Thermo Fisher Scientific). For RAS activation assays, 300 μg of total cell protein was used to pull-down GTP-bound RAS/RAF-RBD complexes according to the manufacturer's instructions (Cytoskeleton). Activated RAS or 20 μg of total cell protein were separated using NuPAGE Novex Bis-Tris gels (Invitrogen/Thermo Fisher Scientific) and transferred to PVDF membrane using an iBlot™ 2 dry blotting system (Invitrogen/Thermo Fisher Scientific). Immunoblot data for this assay were analyzed using the Odyssey® application software v3.030 as described previously (Silva & McMahon, 2014). Statistical significance was determined by *t*-test analyses of three independent experiments.

For immunoblot analysis of pathway expression levels (all blots in Fig 5), a different loading and normalization procedure was followed to allow cell lines to be compared despite reproducible differences in protein extraction efficiency and loading control protein expression. We prepared four replicate samples of each treatment (baseline, stimulation, or steady state) for each of the eight cell lines. Cells were plated in 10 cm plates at a density equivalent to that in our imaging experiments ($1.7 \times 10^6$ cells, ~ 30,000 cells/cm$^2$). Cultures to be stimulated were treated with EGF at 10 ng/ml EGF for 15 min or 2 h, matching the timing of peak activity in live-cell data. Cultures were lysed in 500 μl RIPA buffer with protease and phosphatase inhibitors and clarified as above. These lysates were assayed for protein content by DC protein assay (Bio-Rad) and frozen. Immunoblotting was performed using SDS–PAGE with 12.5% acrylamide or 4–15% gradient gels (Bio-Rad, Cat #4561046, Cat #4561086). To include the many samples in this dataset on the same scale via immunoblot, we loaded lysates based on volume and employed lane-to-lane normalization by total protein load and blot-to-blot normalization by including one sample, the "control", from each blot together on a reference blot. In this way, variations in staining efficiency among membranes were accounted for by scaling all lanes (per target protein) such that the control sample matched its intensity on the reference blot. Normalization to protein load was performed by Ponceau S stain and is applied for all samples prior to normalization across blots, such that variation in the Ponceau stain is also included in the reference blot. Intensity measurements were performed using ImageJ, with background samples collected adjacent to each band/region of interest.

### Phos-Tag immunoblotting

We employed the Phos-Tag method using precast gels (SuperSep Phos-Tag 12.5% Cat #195-17991, and 7.5% Cat #192-18001,

FUJIFILM Wako Chemicals). However, in accordance with previous observations with these gels (Kinoshita-Kikuta *et al*, 2012) and the observation that they likely have excess Phos-Tag reagent, we use samples collected in EDTA-containing RIPA buffer and we performed electrophoresis with Tris-Glycine running buffer (as with above). Phos-Tag fractional phosphorylation measurements are internally controlled and required neither cross-load or cross-blot normalization. See Appendix for detailed Appendix Supplementary Methods.

### Statistical analysis

For all imaging experiments shown, a minimum of 100 cells were imaged and tracked for each condition. Single-cell data points were excluded as outliers if greater than six standard deviations from the dataset mean. For all analyses, at least three independent experimental replicates were performed. Where indicated, single-cell data were normalized to the median value of the PD0325901-treated period. All statistical and computational tasks were performed using MATLAB. Each single-cell trace was normalized to the minimum value in a 1 h window following treatment with 100 nM PD0325901. Baseline values were calculated by taking the mean of the 2 h window prior to stimulation for each cell. The mean was calculated from a 2 h window following treatment with the specified growth factor or vehicle control.

### Statistical analysis: single-cell metrics

Volatility is calculated as the scaled mean absolute derivative, i.e., the sum of the absolute value of the derivative over a 2 h window following stimulation, divided by the mean of the same window. Responders were defined as cells with (i) post-stimulation ERK activity that significantly increased at least 5% compared to the average baseline value, and (ii) a higher maximum derivative at the time of stimulation compared to the baseline region. Response rate was defined as the time from stimulation to the peak ERK activity in each responding cell.

### Statistical analysis: t-tests for single-cell data

Unless otherwise indicated, each statistical comparison was made by *t*-test with unequal variances, and false discovery rate was controlled within each dataset via the Benjamini and Hochberg Step-Up procedure ($\alpha = 0.05$). Where replicates were available at the single-cell level as well as across experiments, the variance of the mean for each experiment was determined from single-cell samples and added to variance across experiments. This corresponds to a linear error model: $\varepsilon_i = \varepsilon_{cell} + \varepsilon_{exp}$, where there error (from the mean) of an individual cell $\varepsilon_i$ equals the sum of the errors arising from cell-to-cell variation $\varepsilon_{cell}$ and from experiment variation $\varepsilon_{exp}$.

## Data availability

All per cell fluorescence data and immunoblot images are provided as source data files for each figure. Note that data presented graphically in Figs 1 and 3 are drawn from the large dataset associated with Fig 2. Raw image data are available via BioStudies: https://www.ebi.ac.uk/biostudies/studies/S-BSST511.

**Expanded View** for this article is available online.

## Acknowledgements

Funding for this work was provided by the National Institute of General Medical Sciences (1R01GM115650 to JGA), the Department of Defense Neurofibromatosis Research Program (W81XWH-16-1-0085 to JGA), and the National Cancer Institute (K01CA197138 to JMS and 1R35CA197709 to FM). Flow-cytometry services were supported by the UC Davis Comprehensive Cancer Center Support Grant (CCSG) awarded by the National Cancer Institute (NCI P30CA093373), and we acknowledge the expert cell sorting assistance of Dr. Bridget McLaughlin and Jonathan Van Dyke. All cell lines were kindly provided by Dom Esposito at the National Cancer Institute Ras Initiative, Frederick, MD.

## Author contributions

MP, JGA, FM: Conceptualized the study; TEG, MP, JMS, FM, JGA: Designed the experiments; TEG, CKT, JMS, FM: Prepared cell lines; TEG, CKT: Performed live-cell experiments; TEG, MP, JMS: Performed immuno blot experiments; MP: Prepared mathematical models; TEG, MP, JMS, JGA: Analyzed the data; TEG, MP, JGA: Wrote the manuscript; TEG, MP, JMS, CT, FM, JGA: Reviewed/edited the manuscript.

## Conflict of interest

The authors acknowledge the following potential sources for conflicts of interest. Frank McCormick is a consultant for the following companies: Aduro Biotech, Amgen, Daiichi Ltd., Ideaya Biosciences, Kura Oncology, Leidos Biomedical Research, Inc., PellePharm, Pfizer Inc., PMV Pharma, Portola Pharmaceuticals, and Quanta Therapeutics. Dr. McCormick has received research grants from Daiichi Sankyo Ltd. and is a recipient of funded research from Gilead Sciences. Dr. McCormick is a consultant and co-founder for the following companies (with ownership interest including stock options): BridgeBio, DNAtrix Inc., Olema Pharmaceuticals, Inc., and Quartz. Dr. McCormick is Scientific Director of the NCI Ras Initiative at Frederick National Laboratory for Cancer Research/Leidos Biomedical Research Inc. John Albeck has received research grants from Kirin Corporation.

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
