## [Review Process File · Molecular Systems Biology]

Oncogenic mutant RAS activity is rescaled by the ERK/MAPK pathway

Taryn Gillies, Michael Pargett, Jillian Silva, Carolyn Teragawa, Frank McCormick, and John G. Albeck
DOI: [10.15252/msb.20209518](https://doi.org/10.15252/msb.20209518)

Corresponding author(s): John G. Albeck (jgalbeck@ucdavis.edu)

Review Timeline:

Submission Date:	17th Feb 20
Editorial Decision:	16th Mar 20
Revision Received:	4th Jul 20
Editorial Decision:	13th Aug 20
Revision Received:	2nd Sep 20
Accepted:	21st Sep 20

Editor: Maria Polychronidou

Transaction Report:

Thank you again for submitting your work to Molecular Systems Biology. We have now heard back from the three referees who agreed to evaluate your study. Overall, the reviewers think that the presented findings seem interesting. They raise however a series of concerns, which we would ask you to address in a revision.

As you will see below, the recommendations of the reviewers are rather clear and I think that there is no need to repeat the points listed below. Please let me know in case you would like to discuss any of the issues raised.

On a more editorial level, we would ask you to address the following.

REFeree REPORTS

Reviewer #1:

In this manuscript, Gillies and coworkers analyze the impact of different RAS mutations on RAF/MEK/ERK signaling. A panel of RAS KO MEFs with re-introduced RAS isoforms, including four oncogenic KRAS mutants, is used that express the EKAR3 reporter to quantify ERK activity by live cell microscopy. The cells were stimulated with various growth factors with a subsequent focus on EGF. Interestingly, the peak ERK activity was not increased in the mutants, but rather the baseline ERK activity was slightly higher in the KRAS mutant cells. By immunoblotting, the changes in feedback regulations were investigated. A simple linear mathematical model was parameterized by the measured protein levels, indicating that external factors play a vital role. By comparing doubly phosphorylated ERK with ERK activity, the influence of ERK phosphatases was quantified.

The manuscript contains a very rich data set and explore a crucial question, i.e. the impact of activating RAS mutations, which are common in human tumors, on ERK activity in vivo. The finding that only the baseline, rather than the peak activation upon stimulation is increased reconciles previous seemingly contradictory reports. The study has been carefully conducted, however, certain aspects require more careful explanations and discussions.

Major comments:

1. The experimental system, especially the EKAR3 probe, requires better explanation. In general, the Materials and Methods section is very short and superficial. Figure EV1 should be included in Figure 1 and explained.
2. The Hras/Nras/Kras-Triple-KO cells are missing in the analyses in Fig. 1 and 2. It would be important to quantify the baseline in ERK activity in absence of any RAS/RAF signaling.
3. Figure 4: In the G12C and the Q61R mutant, also the induction of EGFR phosphorylation is significantly reduced compared to KRAS WT cells. Could the authors speculate why this is the case?
4. Figure 4A: only the phosphorylated, but not the total levels of the signaling proteins are shown. However, as can be observed from the immunoblots in Fig. 5, the total protein levels are remarkably different. Therefore, the re-probes for total protein levels should be shown in Fig. 4A and in Fig. 5D. Furthermore, different from what is shown in the quantification and discussed in the text, in Fig.4A the G12C mutant looks more like WT and only the Q61R mutant appears to lack feedback regulation. This needs to be clarified.
4. Fig. 5: As stated above, total proteins vary substantially in the cell lines analyzed. Especially, the HRAS WT cells harbor low protein levels for all components analyzed. Is there supportive evidence that this does not correspond to lower protein levels in general? The reason for these different protein levels should also be discussed.
5. In the second last paragraph of the discussion, the authors speculate that the ratio between long-term (such as Fra-1) and short-term (such as c-Fos) genes may be informative and reflect the increased baseline ERK activity induced by RAS mutants. This hypothesis should be tested in the cell lines analyzed by measurement of such genes.

Minor comments:

1. The nomenclature of the cell lines by one-letter-code in Fig. 1D, 3K-M and 5B-D is confusing. The full names should be indicated. Further, it is totally unclear why only a selection is shown.
2. The Pargett et al. 2017 reference is missing in the bibliography.

Reviewer #2:

The manuscript describes how different Ras mutations that are among the most prevalent in cancer impact on ERK signaling. While the commonly accepted dogma is that mutated Ras proteins robustly activate ERK to induce an oncogenic proliferation phenotype, it has been found that some of the different Ras mutant isoforms actually even can lead to decreased ERK activity, while still inducing an oncogenic phenotype. In the present manuscript, the authors take fibroblast KO cells for different Ras isoforms, and reconstitute them with the wt alleles or cancer-relevant Ras mutated alleles. This allows for a very clean dissection of the effects of these oncogenic Ras alleles (that would be difficult in cancer patient cells that might harbor multiple oncogenic mutations). Using a carefully calibrated ERK FRET biosensor, they measure the single cell ERK responses to different growth factors. Surprisingly, they find that Ras mutations upregulate basal ERK activity to a rather moderate effect. They also show that growth factor stimulation still can activate ERK activity transients (in response to EGF or other growth factors) or even sustained ERK activity patterns (in response to FGF). So Ras mutations do not upregulate the ERK activity output robustly as commonly believed, and growth factor stimulation pretty much lead to characteristic dynamic

ERK activity profiles. That means that the ERK pathway can rescale in presence of constitutive activation of one of its signaling node. However, the fidelity of the signaling responses (eg the number of cells that can produce a signaling response) is affected.

They then measure a comprehensive number of signaling nodes in response to mild ERK perturbation. This allows them to study negative feedback strengths to upstream signaling nodes in the pathway. Intriguingly, they find that the K-Ras mutation leads to lower levels of negative feedback to upper signaling nodes. Using modelling they show that not only negative feedback but also additional factors are needed for the rescaling mechanism. They propose that this involves phosphatases that can fine tune ERK activity. In the discussion, they comment on this new complexity in the Ras-MAPK network that allows rescaling of the ERK output, and how this might be important in cancer.

I really enjoyed reading this excellent paper, and think it is a very good candidate for publication in MSB. I am ready to accept it with some revisions.

Major comments:

I think further controls are necessary for Figure 6A-B.

- The change in slopes in the regression curves between baseline and Peak or Steady State condition might suggest a saturation of the ERK reporter or a non-linearity of it at high activity levels, leading to misinterpretation in the ratio showed in 6B. Could the authors comment on that? Did the author observe the same by phospho-immunoblot with an endogenous ERK substrate?
- The very high ppERK/ERK activity ratio for KRAS-WT in baseline condition is to me the result to a very low ERK activity measurement. In this extreme condition, again, the FRET reporter could not work linearly and mislead to a very high phosphatase activity. The authors should be more cautious in interpreting this result.
- A dose response curve with a MEK inhibitor would be an interesting experiment to see if there is a loss of linearity in ppERK/ERK activity ratio at very high or very low levels.

Minor comments:

1. The authors make a strong argument that K-, N- and H-Ras are the only Ras GTPases in their system. They expect a BRAF mutant not to have an effect in their system because of absence of the three GTPases. However, BRAF expression still leads to some expression. I think the authors have ignored R-Ras, another Ras GTPase isoform that could still kick in the regulation of BRAF.
2. Have the authors tested that Ras isoforms are expressed at identical levels than in a parental cell lines. There are numerous stories that overexpressed Ras proteins display stronger phenotypes than knockin mutations.
3. There is quite some modelling in the paper, but it is confined in the supplementary figures and text. I think it would add to the strength of the paper, if the modelling approach was schematized in a main figure of the paper.

Reviewer #3:

In their manuscript entitled "Oncogenic mutant RAS signaling activity is rescaled by the ERK/MAPK pathway", Albeck and colleagues describe a study where they systematically profiled signaling in a set of isogenic cell lines containing different WT and mutant RAS proteins. In particular, they make use of a ERK activity FRET sensor as well as immunoblotting, and profile the response of these cell lines with and without perturbation by different ligands. The main results are: ERK activity is only slightly elevated in RAS mutant cell lines, and overall responsiveness of these cell lines seems to be attenuated. Furthermore, the differences seem not necessary to correlate with expression levels of pathway members, and also feedback seems to play a minor role, but seem to correlate with phosphatase activities.

Overall, I think this is an interesting and important study and seems to be carefully done in many regards. But I have a number of major concerns:

1) expression level analysis

I appreciate the effort to characterize the relative expression levels of the major players in the MAPK cascade across cell lines, however I am unsure if the changes are rather artefacts. For instance, we see a decrease in MEK levels post stimulation (Fig. 5B, Total MEK1/2), but MEK is known to be a very stable, long lived protein. Is MEK strongly degraded after phosphorylation? Would be a major finding, but then this would require additional experiments. I also strongly appreciate that the authors show the original blots in the supplement, which unveils the major problem: For instance, the 2hour time points are on a separate blot (on which MEK was generally rather lowly detected). The strategy to normalize by one control lysates then leads to systematic shifts for this condition. As this is also not a major finding and the expression analysis leads to a negative result, I would actually remove this data OR be more thoroughly in quantification.

2 Clonal effects

Clearly, there are differences in ERK activity between WT and mutant lines, but I am not so sure what to make of the differences between isoforms. While it is great to do studies on isogenic cell lines, it is very difficult to assess how much of the variability is due to clonal effects, and how much actually through the transgene. Although much work, I think it is essential to profile clones with the same mutation (maybe 3 WT, 3 mutations), only then one can appreciate those differences.

3 Data for the BRAF_V600E cell line

The authors show that cells expressing BRAF_V600E do not show strongly elevated ERK activity. This is contrary to all studies that I know, which basically show that BRAF V600E expression results in a huge increase in pERK (as BRAFV600E is catalytically active and not responsive to feedback regulation), and cells are subsequently stressed and entering senescence/apoptosis. Now I suspect that when establishing these cells, one selects cells that somewhat gate ERK activity and reduce it to a level that cells can survive. Any evidence for that? How else can one understand that BRAF only activates ERK mildly? How is MEK phosphorylated in these cells?

4 Direct evidence for phosphatase activity

A major (important) result of the manuscript is that phosphatases are essential to buffer ERK activity, and may play a role to buffer ERK and MEK phosphorylation in response to KRAS mutation. I think this would need to be a bit more substantiated. One can directly assess phosphatase activity for e.g. ERK or ERK targets, by blocking MEK or ERK pharmacologically and measure dephosphorylation kinetics by immunoblotting. I think the authors should do it, maybe not for all, but for extreme cases (For example KRAS WT vs one KRAS mutant). Also, I would like to see transcriptomes for these cell lines that show how expression levels of e.g. phosphatases at baseline differ between cell lines, and how different the clones are anyway (see also 1)

More minor

In Figure 1E, it remains unclear which cell lines and conditions have actually been used. Please indicate the cell lines, and which point is which cell line and condition.

Feedback: Saturation of assays?

In Fig. 4 the authors show that feedback is attenuated in RAS mutant cells, which is somewhat counterintuitive. But then they end this section that one cannot rule out saturation of assays (I guess this mainly refers to the pMEK blots). If this is so, why not repeat the assays with reduced proteins?

In Figure 5. Representative Western blots are shown. However, as each graph displays multiple experiments, what do these Western blots actually show? Baseline?

In section "RAS mutants only...":

Volatility is defined as mean absolute derivative normalized to the mean, and then introduced as time-dependent COV. I have three problems with this: First, Trivially cells with higher mean have lower volatility (as one divides by the mean), and also I guess cells with higher mean have less variability in the measurements. Second, I don't see a formal connection with COV. Third, it is nowhere formally defined how it is computed. So please either explain it more clearly, and show how the derivative scales with the mean in the data, or remove it.

The labeling of the cell lines is changing throughout the manuscript. Sometimes it is HRAS WT, sometimes it is H. It should be consistent across figures. I prefer HRAS WT.

Response to reviews**MSB-20-9518****Oncogenic mutant RAS signaling activity is rescaled by the ERK/MAPK pathway**

We thank the reviewers for providing very constructive and helpful comments, which we believe have helped to improve the manuscript substantially. While some of the critiques suggest additional experiments, we have had to be highly selective in choosing which points could be addressed experimentally, given the continuing circumstances that limit capacity for lab work. In some cases, we have explained why we feel that certain experiments, while reasonable suggestions, are not strictly necessary to support the conclusions of the manuscript. We hope that these explanations do not appear dismissive of the reviewers' supportive evaluations, which we value and appreciate greatly.

Reviewer #1:

In this manuscript, Gillies and coworkers analyze the impact of different RAS mutations on RAF/MEK/ERK signaling. A panel of RAS KO MEFs with re-introduced RAS isoforms, including four oncogenic KRAS mutants, is used that express the EKAR3 reporter to quantify ERK activity by live cell microscopy. The cells were stimulated with various growth factors with a subsequent focus on EGF. Interestingly, the peak ERK activity was not increased in the mutants, but rather the baseline ERK activity was slightly higher in the KRAS mutant cells. By immunoblotting, the changes in feedback regulations were investigated. A simple linear mathematical model was parameterized by the measured protein levels, indicating that external factors play a vital role. By comparing doubly phosphorylated ERK with ERK activity, the influence of ERK phosphatases was quantified.

The manuscript contains a very rich data set and explore a crucial question, i.e. the impact of activating RAS mutations, which are common in human tumors, on ERK activity in vivo. The finding that only the baseline, rather than the peak activation upon stimulation is increased reconciles previous seemingly contradictory reports. The study has been carefully conducted, however, certain aspects require more careful explanations and discussions.

Major comments:

1. The experimental system, especially the EKAR3 probe, requires better explanation. In general, the Materials and Methods section is very short and superficial. Figure EV1 should be included in Figure 1 and explained.

We thank the reviewer for this suggestion, and have incorporated figure EV1 into Figure 1. We have added the following additional explanation in the text regarding the mechanism of the EKAR3 reporter (line 105):

“EKAR3 is directly phosphorylated by ERK, acting as a synthetic substrate. Intramolecular binding of the reporter's WW domain to the phosphorylated residue in the substrate domain induces a FRET interaction that can be visualized by observing changes in the CFP/YFP ratio using time-lapse fluorescence microscopy. This interaction is reversible by phosphatases, allowing the reporter to indicate transient changes in the ERK:phosphatase activity ratio (Fig. 1A).”

Additional detail has also been added to the Figure legend (line 727).

2. The Hras/Nras/Kras-Triple-KO cells are missing in the analyses in Fig. 1 and 2. It would be important to quantify the baseline in ERK activity in absence of any RAS/RAF signaling.

We agree that this measurement would make for a more complete dataset, but we think that it is not essential to supporting the paper's conclusions, and it is experimentally quite difficult to collect. The HRAS/NRAS/KRAS triple knockout cells do not proliferate (PMID 20150892), which makes it difficult to set up a reporter cell line for this genotype. We could in principle develop a reporter line in the parental HRas^{-/-}, NRas^{-/-}, KRas^{lox/lox} cell line, and then execute the knockout of the final KRas allele and collect data on these quiescent cells. We expect that performing this experiment adequately would take at least 3 months once our labs are back up and running. Presumably, the potential range for this activity measurement lies between that of the unstimulated KRas cells and the fully MEK-inhibited cells, which is relatively small, as shown in Fig. 4A. This measurement would not add substantially to the main comparison that the paper seeks to make between the wild type and mutant forms of Ras. Thus, we would respectfully argue that the result of this experiment does not justify the amount of time it would take.

3. Figure 4: In the G12C and the Q61R mutant, also the induction of EGFR phosphorylation is significantly reduced compared to KRAS WT cells. Could the authors speculate why this is the case?

The limited increase in pEGFR seen in mutant cells may be due to a lower density of EGFR receptors in the G12C and Q61R mutant cells. ERK activation is known to increase receptor internalization and decrease receptor expression, and thus the elevated levels of baseline ERK seen in the mutant lines may lead to a decrease in receptors at the cell surface. We have added the following to the text (line 236).

“This reduced pEGFR response is potentially due to increased receptor internalization stimulated by chronic ERK activation in mutant cells, resulting in a lower density of receptors on the cell surface.”

4. Figure 4A: only the phosphorylated, but not the total levels of the signaling proteins are shown. However, as can be observed from the immunoblots in Fig. 5, the total protein levels are remarkably different. Therefore, the re-probes for total protein levels should be shown in Fig. 4A and in Fig. 5D. Furthermore, different from what is shown in the quantification and discussed in the text, in Fig.4A the G12C mutant looks more like WT and only the Q61R mutant appears to lack feedback regulation. This needs to be clarified.

It is true that protein levels vary significantly between the cell lines. However, all direct comparisons made in Figure 4A are between different time points within the same cell line – only ratios of these measurements are compared between the cell lines. Our analysis in Figure 5B indicates that the variation in protein levels within cell lines at the time points measured is relatively minor. Because of the way the blots in Figure 4 were run, it is unfortunately not possible to perform a convenient re-probe of the existing samples. However, we think that it is very unlikely that such an experiment would change our conclusions for this figure, for 2 reasons. First, the cell number does not change drastically over the 2-hour period where this analysis was done, and therefore the ppMEK signal (which has been controlled for loading on each blot) reflects the total amount of ppMEK activity per cell, which is more relevant for ERK activation than the ratio of ppMEK/total MEK. Second, even if there were a 2-3 fold change in total MEK only in the mutant cells (which is not supported by the total protein analysis in Figure 5B), the ERKi-mediated increase in ppMEK/total MEK would still be smaller in both mutants

than in the wild type. Because this increase is the indicator of ERK-mediated negative feedback, the conclusion that negative feedback is reduced rather than increased in mutant cells would remain valid.

We agree that there are some differences between the G12C and Q61R mutant lines, and we have updated the text to make this clearer. However, the key indicator of feedback regulation on the pathway is the increase in phosphorylated protein levels that results upon treatment with ERKi. These values are clearly decreased for both mutant cell lines. We have also updated the text to clarify this. The new paragraph reads as follows (line 233):

“However, a different pattern was observed in stimulated KRAS^{G12C} and KRAS^{Q61R} cells. While pEGFR, pAKT, and ppMEK were all significantly increased by EGF stimulation, no significant increase was detected in RAF-RBD pulldown of RAS for either mutant (Fig. 4A,B). The increases in pEGFR and ppMEK, though significant, were lower in magnitude in both KRAS^{G12C} and KRAS^{Q61R} compared with KRAS^{WT}. This reduced pEGFR response is potentially due to increased receptor internalization stimulated by chronic ERK activation in mutant cells, resulting in a lower density of receptors on the cell surface. The EGF-stimulated increase in ppERK in KRAS^{G12C} cells was similar to KRAS^{WT} cells, but reduced in KRAS^{Q61R} cells, consistent with EGF-stimulated ERK activity measurements for these cell lines. As in the KRAS^{WT} cells, negative feedback was assessed by the increase in EGF-stimulated phosphorylation of each protein in the presence of ERKi relative to the vehicle treatment. Unlike KRAS^{WT} cells, pEGFR and RAF-RBD-bound RAS were not further increased by ERKi. Similarly, ppMEK and pAKT were increased by ERKi treatment to a much lesser degree in KRAS^{G12C} and KRAS^{Q61R} cells than in KRAS^{WT}. These data suggest that ERK-mediated feedback is weaker in mutant cells relative to wild type, although they do not rule out the possibility that in the mutant cells, one or more steps in the pathway reaches saturation under these conditions, limiting the ERKi-driven increase.”

4. Fig. 5: As stated above, total proteins vary substantially in the cell lines analyzed. Especially, the HRAS WT cells harbor low protein levels for all components analyzed. Is there supportive evidence that this does not correspond to lower protein levels in general? The reason for these different protein levels should also be discussed.

We now realize that the way we have displayed the western blot data in Figure 5 overemphasizes the differences between these cell lines. All protein level measurements in this figure were corrected for differences in total protein between cell lines by Ponceau staining for the whole lane, which is the accepted best practice for normalization. We chose not to show the total Ponceau stain as it is not easily shown in compact form, but without this measurement, it is not possible to interpret the band darkness; our intention in showing the blots was simply to provide an example image. The data shown in the plots are normalized for the total protein measurement and were meant to provide the relative comparison. To avoid this confusion, we have now removed the blot images from this figure. All blot images are still shown in the supplemental figures.

Regarding the HRAS cells, we did in fact observe lower total protein measurements, which appears to be a unique and reproducible feature of this particular cell line. However, because we corrected for total protein in the plot, the lower readings that are observed in HRAS cells for some of proteins are not attributable to the this factor.

5. In the second last paragraph of the discussion, the authors speculate that the ratio between long-term (such as Fra-1) and short-term (such as c-Fos) genes may be informative and reflect the increased baseline ERK activity induced by RAS mutants. This hypothesis should be tested in the cell lines analyzed by measurement of such genes.

In our previous work on Fra-1 (Gillies et al 2017), and in a recently accepted manuscript (doi: 10.1101/466656) we have performed an extensive investigation of the relationship between ERK activity and Fra-1 and cFos expression. While it would be ideal to repeat these measurements in the MEF cells, we have not yet been able to perform these measurements adequately. However, we think that these papers, along with work from others that we have cited, establish this concept in multiple cell lines with sufficient confidence to include this speculation in the discussion.

Minor comments:

1. The nomenclature of the cell lines by one-letter-code in Fig. 1D, 3K-M and 5B-D is confusing. The full names should be indicated. Further, it is totally unclear why only a selection is shown.

We have added the full names to these figures. For the analysis in Figure 1D, we used this subset of the cell lines because they were sufficient to cover the full range of potential values EKAR values. We have included this explanation in the main text (line 126) and figure legend (line 734).

2. The Pargett et al. 2017 reference is missing in the bibliography.

We have added this reference to the bibliography.

Reviewer #2:

The manuscript describes how different Ras mutations that are among the most prevalent in cancer impact on ERK signaling. While the commonly accepted dogma is that mutated Ras proteins robustly activate ERK to induce an oncogenic proliferation phenotype, it has been found that some of the different Ras mutant isoforms actually even can lead to decreased ERK activity, while still inducing an oncogenic phenotype. In the present manuscript, the authors take fibroblast KO cells for different Ras isoforms, and reconstitute them with the wt alleles or cancer-relevant Ras mutated alleles. This allows for a very clean dissection of the effects of these oncogenic Ras alleles (that would be difficult in cancer patient cells that might harbor multiple oncogenic mutations). Using a carefully calibrated ERK FRET biosensor, they measure the single cell ERK responses to different growth factors. Surprisingly, they find that Ras mutations upregulate basal ERK activity to a rather moderate effect. They also show that growth factor stimulation still can activate ERK activity transients (in response to EGF or other growth factors) or even sustained ERK activity patterns (in response to FGF). So Ras mutations do not upregulate the ERK activity output robustly as commonly believed, and growth factor stimulation pretty much lead to characteristic dynamic ERK activity profiles. That means that the ERK pathway can rescale in presence of constitutive activation of one of its signaling node. However, the fidelity of the signaling responses (eg the number of cells that can produce a signaling response) is affected.

They then measure a comprehensive number of signaling nodes in response to mild ERK perturbation. This allows them to study negative feedback strengths to upstream signaling nodes in the pathway. Intriguingly, they find that the K-Ras mutation leads to lower levels of

negative feedback to upper signaling nodes. Using modelling they show that not only negative feedback but also additional factors are needed for the rescaling mechanism. They propose that this involves phosphatases that can fine tune ERK activity. In the discussion, they comment on this new complexity in the Ras-MAPK network that allows rescaling of the ERK output, and how this might be important in cancer.

I really enjoyed reading this excellent paper, and think it is a very good candidate for publication in MSB. I am ready to accept it with some revisions.

Major comments:

I think further controls are necessary for Figure 6A-B.

- The change in slopes in the regression curves between baseline and Peak or Steady State condition might suggest a saturation of the ERK reporter or a non-linearity of it at high activity levels, leading to misinterpretation in the ratio showed in 6B. Could the authors comment on that? Did the author observe the same by phospho-immunoblot with an endogenous ERK substrate?

Our reporter calibration data (shown in Figure 1D-E) indicate that these measurements are within the linear range of the EKAR3 reporter, even at the highest ERK activities levels seen in EGF-stimulated KRAS^{WT} cells. We added the following note to the text to indicate this (line 319):

“We therefore inferred how this phosphatase activity varied by examining the correspondence between ppERK and activity measurements (Fig. 6A) and estimating the relative phosphatase activity as the ratio of these values (Fig. 6B). All data used for these ratios fell within the linear range for the respective measurements (see Figs. 1E and EV1).”

We attempted within our available capacity to measure the phosphorylation of several endogenous ERK substrates, however we were unable to find a suitable antibody pair for measuring the phosphorylation and total expression of an endogenous substrate in MEF cells. Nonetheless, we agree that additional analysis is needed on this point. To strengthen our conclusion, we now have added figures 6C-G, which support our conclusion of dynamic differences in phosphatase activity. While we were unable to find suitable antibody pairs to assess ERK target phosphorylation relative to total, we were able to calculate the decay rate of reporter phosphorylation after MEK inhibition (6C), which confirms that there consistent differences between the rate of phosphatase activation between baseline and steady state conditions. Additionally, we show that the protein abundance of one candidate phosphatase, DUSP6, is dynamically regulated by ERK activity with a time pattern consistent with the inferred changes in phosphatase activity (Fig. 6D).

- The very high ppERK/ERK activity ratio for KRAS-WT in baseline condition is to me the result to a very low ERK activity measurement. In this extreme condition, again, the FRET reporter could not work linearly and mislead to a very high phosphatase activity. The authors should be more cautious in interpreting this result.

We agree that this measurement is likely an outlier. We have emphasized this in the text and have edited the text to clarify that our focus in this analysis is on the changes in phosphatase activity that occur upon stimulation, which occur in multiple cell lines and do not rely on this particular measurement (line 323):

“At baseline, phosphatase activity appears nearly uniform across cell lines, except for the KRAS^{WT} measurement, which is likely an outlier (Fig. 6B). This consistency in phosphatase activity between cell lines results in a significant linear correlation between ppERK and ERK activity, as expected (Fig. 6A). However, after stimulation, both the slope and correlation are diminished, and variance in the estimated phosphatase activity increases. By the steady state time point, estimated phosphatase activity rises significantly, as the different RAS cell lines settle to very similar levels of ERK activity (Fig. 3L) despite varying levels of ppERK (Fig. 5D). This correlational analysis implies that after stimulation, phosphatase activities and/or levels are regulated in such a manner that they act to normalize the levels of ERK activity, despite residual differences in concentration of ppERK. However, the observation that apparent phosphatase activity at steady state is uncorrelated to ppERK at baseline implies dynamic complexity beyond simple regulation by ERK.”

- A dose response curve with a MEK inhibitor would be an interesting experiment to see if there is a loss of linearity in ppERK/ERK activity ratio at very high or very low levels.

As noted above, our reporter calibration data (Figure 1D-E) show that all of our measurements are within the linear range of the EKAR sensor. The proposed experiment would be useful insofar as addressing the relationship between ppERK/ERK activity in the context of MEK inhibitor, but as the blot we have added in Fig. 6G shows, the addition of this inhibitor is likely to influence phosphatase regulation. Thus, while we appreciate the suggestion from the reviewer, we think this experiment would not necessarily provide additional support for the conclusions in this work.

Minor comments:

1. The authors make a strong argument that K-, N- and H-Ras are the only Ras GTPases in their system. They expect a BRAF mutant not to have an effect in their system because of absence of the three GTPases. However, BRAF expression still leads to some expression. I think the authors have ignored R-Ras, another Ras GTPase isoform that could still kick in the regulation of BRAF.

We agree that R-Ras could be responsible for the observed increase in ERK activity in response to some of the growth factors. We have added text to note this possibility in the case of the ERK responses seen in the BRAF^{V600E} cell line (line 148).

“However, both FGF and IGF induced elevated ERK activity in BRAFV600E cells (Fig. 2C), indicating an ERK response that is not mediated via HRAS, KRAS, or NRAS, which could occur through other GTPases with the potential to activate RAF, such as R-RAS.”

2. Have the authors tested that Ras isoforms are expressed at identical levels than in a parental cell lines. There are numerous stories that overexpressed Ras proteins display stronger phenotypes than knockin mutations.

This question is difficult to address directly. Because the parental MEF cell line has the genotype of HRas^{-/-}, NRas^{-/-}, KRas^{lox/lox}, it does not provide a good control for total RAS expression levels. A potentially useful comparison could be NIH-3T3 cells, but this could also be misleading due to the fact that these cells have been independently immortalized and passaged separately for years.

Nonetheless, although RAS expression is difficult to compare to “normal” cells, our data show that for the most important comparison relevant to our conclusions (Fig. 5B), the wild type and mutant Ras proteins are all expressed at similar levels (except for G12C, which is higher). Given that our data indicate a *smaller* than expected difference in ERK activity between wild type and mutant RAS, the main concern for overexpression would be that we had expressed wild type RAS in excess of the mutants, but this is clearly not the case. The excessive expression of G12C provides a strong counter-argument to this possibility, showing that even with high levels of mutant Ras expression there is a limited increase over wild type signaling levels.

3. There is quite some modelling in the paper, but it is confined in the supplementary figures and text. I think it would add to the strength of the paper, if the modelling approach was schematized in a main figure of the paper.

We agree that this would be a helpful addition to the main figures, and we have added additional details to the schematic in Figure 5A to clarify the modeling approach.

Reviewer #3:

In their manuscript entitled "Oncogenic mutant RAS signaling activity is rescaled by the ERK/MAPK pathway", Albeck and colleagues describe a study where they systematically profiled signaling in a set of isogenic cell lines containing different WT and mutant RAS proteins. In particular, they make use of a ERK activity FRET sensor as well as immunoblotting, and profile the response of these cell lines with and without perturbation by different ligands. The main results are: ERK activity is only slightly elevated in RAS mutant cell lines, and overall responsiveness of these cell lines seems to be attenuated. Furthermore, the differences seem not necessary to correlate with expression levels of pathway members, and also feedback seems to play a minor role, but seem to correlate with phosphatase activities.

Overall, I think this is an interesting and important study and seems to be carefully done in many regards. But I have a number of major concerns:

1) expression level analysis

I appreciate the effort to characterize the relative expression levels of the major players in the MAPK cascade across cell lines, however I am unsure if the changes are rather artefacts. For instance, we see a decrease in MEK levels post stimulation (Fig. 5B, Total MEK1/2), but MEK is known to be a very stable, long lived protein. Is MEK strongly degraded after phosphorylation? Would be a major finding, but then this would require additional experiments. I also strongly appreciate that the authors show the original blots in the supplement, which unveils the major problem: For instance, the 2hour time points are on a separate blot (on which MEK was generally rather lowly detected). The strategy to normalize by one control lysates then leads to systematic shifts for this condition. As this is also not a major finding and the expression analysis leads to a negative result, I would actually remove this data OR be more thoroughly in quantification.

This critique of the variation in immunoblot data is accurate, and we agree that it is more likely due to technical variation rather than an actual change in MEK abundance. Our normalization using the control lysate accounts for differences in blot preparation (transfer, staining, imaging, etc) since the same exact sample is run on all blots. However, it does not account for

experimental noise resulting from differences in sample collection and preparation, which could account for the decreased MEK levels seen in the steady state time point data.

However, while by eye there is a decrease in MEK levels in the steady state condition, we did not find this difference to be statistically significant, and we do not base any conclusions on this difference. In the modeling approach in which the expression data are used, the protein levels of MEK are based on the mean of the samples under all conditions (i.e., the black bars in Fig. 5D), which we think gives the best overall estimate of their abundance. Thus, any spurious variation due to sample or blot differences is averaged out to the best of our ability by including data from multiple conditions, experiments and blots. Given these considerations, we think it is appropriate to continue to include all the data collected in our analysis.

2 Clonal effects

Clearly, there are differences in ERK activity between WT and mutant lines, but I am not so sure what to make of the differences between isoforms. While it is great to do studies on isogenic cell lines, it is very difficult to assess how much of the variability is due to clonal effects, and how much actually through the transgene. Although much work, I think it is essential to profile clones with the same mutation (maybe 3 WT, 3 mutations), only then one can appreciate those differences.

This is a very reasonable critique of our approach, and also a limitation of essentially all studies in which RAS expression or activity are manipulated. We absolutely acknowledge that clonal effects may come in to play here, and we provide a substantial discussion of this caveat in detail in the Discussion (beginning line 411). However, the amount of work needed to address this question would take at least a year to complete and is infeasible for this manuscript. We do plan to follow up on this question in additional studies, which will be carefully designed to allow us to monitor changes in expression level and selective pressures as cells adapt to changes in RAS genotypes.

We think it is worth noting that the current set of samples – 3 wild type and 4 mutant alleles of RAS – already provides strong multi-clonal support for our main conclusions that mutant RAS drives a higher baseline but limited peak ERK activity, relative to wild type RAS alleles. While we do describe the differences between the various mutants, the overall conclusion of the paper deals with mutant vs. WT in general and does not depend critically upon these allele variations. Therefore, while it would certainly be useful to have multiple clones for each isoform to confirm the specific differences between alleles, this level of detail is not needed to support the conclusions that we draw in the current study.

3 Data for the BRAF_V600E cell line

The authors show that cells expressing BRAF_V600E do not show strongly elevated ERK activity. This is contrary to all studies that I know, which basically show that BRAF V600E expression results in a huge increase in pERK (as BRAFV600E is catalytically active and not responsive to feedback regulation), and cells are subsequently stressed and entering senescence/apoptosis. Now I suspect that when establishing these cells, one selects cells that somewhat gate ERK activity and reduce it to a level that cells can survive. Any evidence for that? How else can one understand that BRAF only activates ERK mildly? How is MEK phosphorylated in these cells?

It is certainly true that BRAF^{V600E} often drives strong ERK activity when overexpressed. However, when expressed at endogenous levels, BRAF^{V600E} has been observed to induce more moderate ERK activation (less than growth factor stimulation) shortly after Cre-mediated induction in MEFs (PMID: 16357158; See Fig. 5). Our results are consistent with these findings. Therefore, the moderate level of ERK activity observed in our BRAF^{V600E} cells is not necessarily the result of a very stringent selection or adaptation process. We have added this reference to the Results section (line 147) to help explain what can seem like a surprising result.

Nonetheless, this point is certainly a valid critique that is potentially applicable to many experiments in the literature, and we think our paper helps to bring this potential issue into greater focus. We conclude that multiple processes seem to collaborate to maintain ERK activity within a limited range, and we fully agree that some of these processes may include long-term selective or adaptive effects. The presence of such long-term processes, while complicating, would not invalidate our analysis of other potential mechanisms and would not substantially change our conclusions. As noted in the previous point, we believe an experimental investigation of long-term adaptive effects will certainly be interesting but would be out of the scope for the current manuscript.

4 Direct evidence for phosphatase activity

A major (important) result of the manuscript is that phosphatases are essential to buffer ERK activity, and may play a role to buffer ERK and MEK phosphorylation in response to KRAS mutation. I think this would need to be a bit more substantiated. One can directly assess phosphatase activity for e.g. ERK or ERK targets, by blocking MEK or ERK pharmacologically and measure de-phosphorylation kinetics by immunoblotting. I think the authors should do it, maybe not for all, but for extreme cases (For example KRAS WT vs one KRAS mutant). Also, I would like to see transcriptomes for these cell lines that show how expression levels of e.g. phosphatases at baseline differ between cell lines, and how different the clones are anyway (see also 1)

We agree that this point requires further substantiation. We have now added figures 6C-G, which support our conclusion of dynamic differences in phosphatase activity. While we were unable to find suitable antibody pairs to assess ERK target phosphorylation relative to total, we were able to calculate the decay rate of reporter phosphorylation after MEK inhibition (6C), which confirms that there are consistent differences between the rate of phosphatase activation between baseline and steady state conditions. Additionally, while a complete transcriptome analysis was not possible at this time, we show that a candidate phosphatase, DUSP6, is dynamically regulated in abundance with a time pattern consistent with the inferred changes in phosphatase activity (Fig. 6D). There could certainly be other phosphatases that are dynamically regulated, but this finding, similar to what a transcriptome analysis would provide, confirms that such changes do occur in this system.

More minor

In Figure 1E, it remains unclear which cell lines and conditions have actually been used. Please indicate the cell lines, and which point is which cell line and condition.

We have now added these annotations to the figure.

Feedback: Saturation of assays?

In Fig. 4 the authors show that feedback is attenuated in RAS mutant cells, which is somewhat counterintuitive. But then they end this section that one cannot rule out saturation of assays (I guess this mainly refers to the pMEK blots). If this is so, why not repeat the assays with reduced proteins?

We apologize that this wording was unclear. We intended to point out the possibility of pathway saturation, rather than the saturation of the pMEK assay. We have edited the text to rephrase this (line 244).

“These data suggest that ERK-mediated feedback is weaker in mutant cells relative to wild type, although they do not rule out the possibility that in the mutant cells, one or more steps in the pathway reaches saturation under these conditions, limiting the ERK-driven increase.”

In Figure 5. Representative Western blots are shown. However, as each graph displays multiple experiments, what do these Western blots actually show? Baseline?

As noted in the response to Reviewer 1, we agree that this presentation of the blot images was unhelpful; in addition to showing only one of the conditions, it doesn't account for the normalization process which was performed using Ponceau staining. Our intention in showing the blots was simply to provide an example image. We have now removed the blot images from below these graphs and will simply direct readers to the supplement for blot images in order to avoid the confusion of which condition is shown.

In section "RAS mutants only...":

Volatility is defined as mean absolute derivative normalized to the mean, and then introduced as time-dependent COV. I have three problems with this: First, Trivially cells with higher mean have lower volatility (as one divides by the mean), and also I guess cells with higher mean have less variability in the measurements. Second, I don't see a formal connection with COV. Third, it is nowhere formally defined how it is computed. So please either explain it more clearly, and show how the derivative scales with the mean in the data, or remove it.

We have added additional text to clarify this calculation (line 166):

“To compare the tendency for sporadic and time-varying activity, we sought a metric similar to the coefficient of variation (CV). However, when used on time-series data, the CV neglects time and only reflects how far samples deviate from the mean regardless of when they occurred. We instead compute a metric we term “volatility”. This is calculated by first differentiating the ERK activity per cell, then taking the absolute value and averaging over the time-series. As with the CV, we scale volatility by the mean value for that cell. This metric is the time-dependent equivalent of the CV in that it is the mean-scaled average of deviations from the past time point, where the CV is the mean-scaled average of deviations from the mean.”

The labeling of the cell lines is changing throughout the manuscript. Sometimes it is HRAS WT, sometimes it is H. It should be consistent across figures. I prefer HRAS WT.

We have now included the full cell line names on the graph labels.

Thank you for sending us your revised manuscript. We have now heard back from the three reviewers who were asked to evaluate your study. Overall, the reviewers think that the study has significantly improved as a result of the performed revisions.

As you will see below, reviewer #1 still raises some remaining concerns. Specifically, they think that the lack of quantification of the baseline ERK activity without any RAS/RAF signaling (their major point #2) is problematic, given the focus of the findings on baseline ERK activity. Moreover, they raise technical concerns regarding the heterogeneity of the immunoblotting data shown in Figures 4 and 5. During our pre-decision cross-commenting process (in which the reviewers are given the chance to make additional comments, including on each other's reports), reviewer #2 mentioned: "Reviewer 1 criticizes the approach used by the authors to evaluate the increased ERK activity in mutant cell lines and western blot normalization. The major criticism is on the normalization approach based on Ponceau S staining that could lead to measurement noise in the western blots. Ponceau S is used by a lot of different people for normalizing loadings. The blots have been performed by the lab of Frank McCormick who is one of the early fathers of Ras/MAPK signaling. I would trust their experience in western blotting. We agree with reviewer #1 that there are differences in BRAF, MEK and ERK expression in different cell lines. It is difficult to discern if this comes from technical noise in the WB experiments, or if these differences occur because of adaptation of these cell lines. We believe this does not alter the conclusions of the paper given the evident effects displayed by the Ras mutants, but maybe the authors can simply discuss this in the paper a bit more precisely (which is currently not done). Regarding the general criticism about baseline activity, the reviewer claims that a general baseline for all the cell lines is not provided. However, the authors mention that the triple KO cells in the 3 Ras isoforms are not viable, precluding performing these experiments (including a BRAF expression experiment). In our opinion the entire experimental setting regarding the re-expression of RAS isoforms and their oncogenic mutations are well controlled, not needing further baseline experiments. In our opinion, these criticisms should not hamper the publication of this work." As such, we would like to offer you a chance to address the remaining issues raised by reviewers #1 and #3 in a round of minor revisions. The issue regarding the data heterogeneity can be addressed by including some discussion as reviewer #2 recommends. Experimental quantification of the baseline ERK activity in triple mutant cell lines is not required for the acceptance of the work for publication.

On a more editorial level, we would like to ask you to address the following.

REFEREE REPORTS

Reviewer #1:

In the revised manuscript, which is a major revision, Gillies and coworkers partially responded to the issues raised. They give a better explanation of the EKAR3 probe and speculate on the origin of the reduced EGFR activation in the G12C and Q61R mutants (major comments 1 and 3). They now also further better explain on the consequences of the different total levels of the immunoblots and refer to a previous publication concerning the ratio between ERK-induced long-term and short-term signal activation (major comments 4A and 5). Additionally, the minor comments were addressed by adding the missing reference and using the full names of the cell lines in the main figures. However, the one letter code for the cell lines is still used in Figure EV1 and in the Westernblot Source Data for Figure 5, which makes it extremely difficult to assess the data displayed in these figures. Therefore it is of importance to be consistent and use the full names of the cell lines throughout the manuscript.

Major comment 2 (baseline in ERK signaling) was not addressed and it was stated that the experiment would take too much time and would not significantly advance the manuscript. However, because the authors heavily focus on the baseline of ERK signaling that is enhanced in the KRAS mutants and also observe that upon stimulation with FGF and IGF (and also to some degree with EGF) elevated ERK activity is detectable in BRAFV600E cells, which the authors speculate could be mediated by R-RAS, it is important to quantify the baseline of the observations. A main issue that remains is the heterogeneity of the immunoblotting data, which is a key basis for the mathematical modeling. It appears that some of this heterogeneity could be of technical origin. For example compared to the other cells lines HRAS-WT cells consistently seem to harbor less BRAF, MEK and ERK, which is odd given the information in the methods sections that in each case 20 µg of total protein was loaded. Further, whereas in Figure 4 normalization of the immunoblot data by beta-Actin or Pan RAS was performed, in Figure 5 protein level measurements were corrected for by Ponceau S staining considering the whole lane, which the authors claim is "accepted best practice for normalization" without giving any reference demonstrating the validity of this statement. While staining with Ponceau S is frequently performed for visual inspection to check for equal transfer, the Ponceau S stain is hardly quantitative. For example, washing of the blots after staining has to be performed very carefully and can easily induce artefacts in the staining. It rather appears that this procedure amplifies the variability of the results. Due to the single letter code and the poor alignment of the blots in the "Western Blot Source Data for Figure 5" it is difficult to compare the individual blots but it is frequently visible that there are systematic errors in the ERK signals and the levels of the corresponding Ponceau S stain are not always matching. Additionally, the displayed MEK signals are too weak to conclude if this also holds true for this protein. In general, the "Western Blot Source Data for Figure 5" is not comprehensible and needs to be revised: The labeling is confusing and the blots are not aligned.

Given the heterogeneity of immunoblot quantifications shown in Figure 5, it is striking that in Figure

4B the error of the mean of triplicates is indicated and appears for most determinations rather small. These quantifications support the conclusion of a weakened feedback from ERK, however in the immunoblot data displayed in Figure 4A still the mutant G12C looks more like WT and the reduced activation of MEK (ppMEK) is not visible. As presented, focusing only on the relative increase compared to baseline in each individual cell line, a direct comparison of the extent of activation between cell lines is not possible. Here determination of total levels and an analysis on the same blot would be important. In particular, the requested re-probe of the EGFR levels would be essential to validate the hypothesis of the authors that reduced pEGFR response is due to increased receptor internalization and degradation upon chronic RAS activation. The data displayed in Figure 5B suggests that the results obtained for the cell lines are much more variable compared to the KRAS-WT cell line, which makes firm conclusions difficult. Only for two of the mutant cell lines the elevated basal levels of activated ERK identified by the analysis of the EKAR reporter was confirmed by the immunoblot data in Figure 5B. Therefore, the statement that based on the ERK activity determined by EKAR the possibility that any individual component acts as a limiting factor was ruled out has to be treated with caution. Rather, per cell line the relation of ERK activation to total amount of pathway components has to be better established. In conclusion, while the manuscript was improved by the revision, several issues remain. At present the provided evidence does not support the statement in the abstract "We identify roles for pathway-level effects...that act to rescale pathway sensitivity independent of expression level".

Reviewer #2:

The authors have addressed my concerns. I strongly support publication of this excellent article that is in my view very important for the Ras/MAPK signaling field.

Reviewer #3:

The authors addressed most of my concerns, I particularly value their analysis of the deactivation kinetics. The only thing that remains is that I think the authors should reconsider the idea that DUSP6 dephosphorylates ERK targets. DUSP6 is an MAPK specific phosphatase that gains its specificity by a specific binding domain of ERK (which I doubt are present at ERK targets or the reporter).

Response to Review: MSB-20-9518R

We thank the reviewers for a thoughtful and constructive process, and for the additional comments on the revision, all of which have helped to improve the manuscript.

Reviewer 1

However, the one letter code for the cell lines is still used in Figure EV1 and in the Westernblot Source Data for Figure 5, which makes it extremely difficult to assess the data displayed in these figures. Therefore it is of importance to be consistent and use the full names of the cell lines throughout the manuscript.

We apologize for overlooking this, and we have corrected the labels in these supplemental sections.

Major comment 2 (baseline in ERK signaling) was not addressed and it was stated that the experiment would take too much time and would not significantly advance the manuscript. However, because the authors heavily focus on the baseline of ERK signaling that is enhanced in the KRAS mutants and also observe that upon stimulation with FGF and IGF (and also to some degree with EGF) elevated ERK activity is detectable in BRAFV600E cells, which the authors speculate could be mediated by R-RAS, it is important to quantify the baseline of the observations.

We realized that one of our existing experiments provides insight into this important question. In Figure 2A, we show the time course of KRAS^{G12C} cells treated with an inhibitor (ARS-853) specific for this allele. Because we are inhibiting the only RAS isoform present in these cells, the residual activity can be interpreted as the baseline value for cells lacking RAS activity. To highlight this comparison, we have now included the following text in the Results section (lines 139-143):

“Furthermore, because ARS-853 inhibits the only KRAS isoform present in KRAS^{G12C} cells, we used this condition to estimate the RAS-independent background level of ERK activity. Following ARS-853 treatment, ERK3 signal decreased to a level approximately equivalent to that of untreated KRAS^{WT}, followed by a small rebound. This similarity suggests that the strength of ERK activity contributed by RAS-independent sources is near the minimal baseline value.”

A main issue that remains is the heterogeneity of the immunoblotting data, which is a key basis for the mathematical modeling. It appears that some of this heterogeneity could be of technical origin. For example compared to the other cell lines HRAS-WT cells consistently seem to harbor less BRAF, MEK and ERK, which is odd given the information in the methods sections that in each case 20 µg of total protein was loaded.

We acknowledge that our description of our blotting methods and results did not clearly delineate between the two different approaches used for figures 4 and 5, which is one source of the apparent heterogeneity. Figure 4 was conducted with an approach that is more conventional, in which extracted samples were loaded according to quantified protein amount (20 µg) and normalized by loading control proteins. However, for the cross-cell line comparison in Figure 5, we found that this approach could not be used due to variation in the expression of typical loading control proteins. Moreover, some cell lines, particularly HRAS, consistently showed lower concentrations of extracted proteins, making loading constant total protein impractical for these blots as it would force us to use reduced amounts for the other cell lines. We therefore used a different protocol for Figure 5, in which samples were loaded based on volume (i.e., not 20 µg total protein each) and then normalized by total protein stain for each lane. We believe that this strategy accounts for much of the heterogeneity observed by the reviewer. While our choice of this approach was made to maximize our ability to quantitatively compare between cell lines, we acknowledge that it unfortunately makes it quite difficult to read the blots by eye. Nonetheless, we believe that this method is better suited to the particular needs of the analysis in Figure 5 than a more conventional approach.

We have now made multiple edits to the Results and Methods sections to more clearly describe where the two different approaches were used and specifically how they differed (lines 266-271, line 549, lines 563-566).

Further, whereas in Figure 4 normalization of the immunoblot data by beta-Actin or Pan RAS was performed, in Figure 5 protein level measurements were corrected for by Ponceau S staining considering the whole lane, which the authors claim is "accepted best practice for normalization" without giving any reference demonstrating the validity of this

statement. While staining with Ponceau S is frequently performed for visual inspection to check for equal transfer, the Ponceau S stain is hardly quantitative. For example, washing of the blots after staining has to be performed very carefully and can easily induce artefacts in the staining. It rather appears that this procedure amplifies the variability of the results.

Our statement regarding the best practice was based on the Journal of Biological Chemistry Author guideline website which states:

Normalize signal intensity to total protein loading (assessed by staining membranes for total protein) whenever possible. "House-keeping" proteins should not be used for normalization without evidence that manipulations do not affect expression.

<http://jbcresources.asmb.org/collecting-and-presenting-data#blot>

We recognize that this approach is not common practice for many labs, and admittedly, we do not follow all of JBC's guidelines ourselves. Our intention in making this statement was simply to defend the use of membrane staining as a valid quantitative technique. We did perform this procedure, including the washing and imaging, very carefully, and we believe that it performed as well or better than the alternatives, given the variability between cell lines noted above. The variability that is apparent in the raw data is more likely a result of the cell line differences and the loading scheme described above.

Due to the single letter code and the poor alignment of the blots in the "Western Blot Source Data for Figure 5" it is difficult to compare the individual blots but it is frequently visible that there are systematic errors in the ERK signals and the levels of the corresponding Ponceau S stain are not always matching. Additionally, the displayed MEK signals are too weak to conclude if this also holds true for this protein. In general, the "Western Blot Source Data for Figure 5" is not comprehensible and needs to be revised: The labeling is confusing and the blots are not aligned.

As noted above, our approach to setting up quantitative cross-cell line comparisons makes reading the blots by eye difficult. Thus, the differences in ERK signals noted by the reviewer are not necessarily systematic errors. We have done our best to correct these issues by organizing and re-labeling the blots in the supplement and by describing the methods more clearly.

Given the heterogeneity of immunoblot quantifications shown in Figure 5, it is striking that in Figure 4B the error of the mean of triplicates is indicated and appears for most determinations rather small.

As clarified above, the immunoblot approaches used for figures 4 and 5 were different. The within-cell line comparisons performed in figure 4 showed much less heterogeneity and therefore had lower error estimates.

These quantifications support the conclusion of a weakened feedback from ERK, however in the immunoblot data displayed in Figure 4A still the mutant G12C looks more like WT and the reduced activation of MEK (ppMEK) is not visible. As presented, focusing only on the relative increase compared to baseline in each individual cell line, a direct comparison of the extent of activation between cell lines is not possible. Here determination of total levels and an analysis on the same blot would be important.

We agree that the blots for WT and G12C show a roughly similar pattern by eye. However, we maintain that an adequate assessment of ERK-dependent negative feedback on MEK for each cell line can be made by comparing ppMEK in the absence and presence of ERK inhibitor for that line, as this comparison demonstrates the extent to which ERK activity limits ppMEK activation. While the quantification has been normalized to the baseline for each cell line, the relative values for ppMEK in untreated and ERKi-treated cells can be compared regardless of the baseline value. This comparison depends on two different ratios (WT lane 2/lane 5 vs. G12C lane 2/lane 5), and we think it is difficult to make this assessment reliably by eye. This is why we have organized the bar graphs in figure 4B to show this comparison side by side. To help clarify the comparison that we are making we have added the following sentence on lines 225-227:

“In Figure 4B, the comparison between untreated and ERKi-treated cells is shown side by side (dark- and light-shaded bars, respectively) to emphasize the effect of ERK-mediated feedback for each species.”

In particular, the requested re-probe of the EGFR levels would be essential to validate the hypothesis of the authors that reduced pEGFR response is due to increased receptor internalization and degradation upon chronic RAS activation.

We have removed this hypothesis and have reworded our interpretation of this result more descriptively (lines 243-246):

“The increases in pEGFR and ppMEK, though significant, were lower in magnitude in both KRAS^{G12C} and KRAS^{Q61R} compared with KRAS^{WT}, indicating a decreased pathway responsiveness relative to baseline, which is consistent with the ERK activity measurements for these cell lines (Fig. 3).”

The data displayed in Figure 5B suggests that the results obtained for the cell lines are much more variable compared to the KRAS-WT cell line, which makes firm conclusions difficult. Only for two of the mutant cell lines the elevated basal levels of activated ERK identified by the analysis of the EKAR reporter was confirmed by the immunoblot data in Figure 5B. Therefore, the statement that based on the ERK activity determined by EKAR the possibility that any individual component acts as a limiting factor was ruled out has to be treated with caution. Rather, per cell line the relation of ERK activation to total amount of pathway components has to be better established.

We think it is important to note that the discordance between ERK activity measurements and phospho-ERK immunoblots does not necessarily imply that the immunoblot measurements are unreliable; these values can differ due to different phosphatase activity levels, as shown in Figure 6. Nonetheless, it is of course true that our conclusions depend on the precision of our measurements. While we have carried out this comparison as carefully as possible for an immunoblot approach, we acknowledge that there is room for uncertainty that our measurements conclusively demonstrate a lack of involvement of expression levels. Accordingly, we have reworded our conclusion for this analysis (lines 279-282):

“Second, levels of BRAF, MEK and ERK, but not CRAF, varied significantly among cell lines, but without a clear pattern or correlation structure. When total expression levels were compared with ERK activity (via EKAR), no significant correlations were found, suggesting that no individual component acts as a limiting factor.”

We have also added a comment to the Discussion section (lines 430-432) to note the limits of our analysis:

“Furthermore, while our limited immunoblot-based analysis was unable to identify differences in expression that explain the divergence of ERK signaling from the expected, a more precise and comprehensive proteomic analysis could reveal overlooked correlations (Shi et al., 2016).”

In conclusion, while the manuscript was improved by the revision, several issues remain. At present the provided evidence does not support the statement in the abstract “We identify roles for pathway-level effects...that act to rescale pathway sensitivity independent of expression level”.

Given the uncertainty involved in fully accounting for expression differences, we agree that more caution is needed for this conclusion. We have removed the phrase “independent of expression level” from this sentence in the abstract (line 19).

Reviewer 3

The only thing that remains is that I think the authors should reconsider the idea that DUSP6 dephosphorylates ERK targets. DUSP6 is an MAPK specific phosphatase that gains its specificity by a specific binding domain of ERK (which I doubt are present at ERK targets or the reporter).

We agree that it is unlikely for DUSP6 to act directly on the reporter. We intended to say that the observed changes in DUSP6 imply that phosphatase expression can change on a time scale consistent with the inferred phosphatase activity. We have reworded our interpretation as follows (lines 361-364):

“While DUSP6 is known to be primarily specific for ERK due to a direct binding interaction, its dynamic regulation demonstrates that phosphatase expression can shift on a time scale consistent with the inferred phosphatase activity. Additional work will be needed to more clearly identify the phosphatases that act on EKAR or on endogenous ERK substrates.”

Thank you again for sending us your revised manuscript and for making the final requested changes. We are now satisfied with the modifications made and I am pleased to inform you that your paper has been accepted for publication.

Corresponding Author Name: John Albeck

Manuscript Number: MSB-20-9518R